

# Identification of early warning criteria for rough sea ship navigation using high-resolution numerical wave simulation and shipboard measurements

Chen Chen[1], Kenji Sasa[1], Takaaki Mizojiri[2]

[1] Department of Maritime Sciences, Kobe University, Kobe, Japan
    5-1-1, Fukae Minami Machi, Higashinada-Ku, Kobe, 658-0022 JAPAN
[2] Imabari Shipbuilding Co. Ltd., Japan

*Correspondence to*: Chen Chen (cc198895@hotmail.com)

**Abstract.** The analysis of ocean surface waves are essential to ensure a safe and economical navigation. Since 2010, different on-board observation data from a bulk carrier have been collected for 6 years, including high-risk shipping regions in the Southern Hemisphere with strong ocean currents. For four rough sea cases, high-resolution numerical simulations of ocean waves, including the effect of wave-current interaction on ship navigation, have been performed using the WAVEWATCH III model. The simulations considered the ocean surface wind force from the widely used grid point value database NCEP-FNL and ERA-Interim. Aimed at providing practical suggestions for safe navigation by avoiding possible high-risk ocean regions as well as the construction of a more effective and efficient optimum ship routing system, the model results were validated based on on-board observations, followed by discussions on the responses of ship motion and navigation to wave states at different levels. Finally, identification of the early warning criteria, including various operational ocean parameters, is provided for ballast and loaded ships sailing in rough seas.

## 1 Introduction

Marine transportation delivers 90% of all the cargo worldwide, making it a significantly important factor for human life. Due to an increase in ship size and number, shipping activities frequently lead to a higher possibility of ship accidents and increased safety risk to human beings, property losses, and the pollution of ocean environments. Therefore, marine weather information,

including an accurate forecast of extreme ocean surface wave states. Therefore, the analysis of the effects of waves on ship



navigation is essential for safe, economical, and environment-friendly ship navigation, from the viewpoint of ship weather routing.

Statistical analysis of ship accident database has been used as an important method by several researchers to find out the relationship between sea states and ship safety as well as make identification of warning criteria.

(Toffoli et al., 2005) studied ocean state parameters and ship characteristics from 270 ship accidents caused by bad weather. They aimed at contributing to the identification of adequate warning criteria for safe navigation. Through their analysis of the sea states during the ship accidents, they found that sea state parameters such as the significant wave height, wave steepness, crossing seas and rapid development of the sea states were important. For example, they gave statistical statements that 2/3 of all analysed ship accidents happened in a sea state with the significant wave height lower than 4 m, 3/5 of them with wave

steepness between 0.0300 and 0.0450, 1/2 of them were observed in crossing seas with a direction range of wave trains such as swell and wind seas at ±30°. They also mentioned that a rapid increase in sea states was quite often observed in their investigation including almost 4/5 of all events. Moreover, they argued that due to the coarse resolution of the wave model, sea states were apparently underestimated, especially for the strong tropical storms and wave-current interaction effects. The information on the exact time and ship characteristics of each accident was not available, thus leading to limitations of a more

detailed and accurate discussion.

In their review study on the wave forecasts and small vessel safety, (Niclasen et al., 2010) focused on sea state parameters that are important for small vessel safety; they tried to find high-risk situations that were not resolved by traditional wave parameters such as wave height, wave period, and direction. They found that a combination of wave height and steepness or the calculation of risk of the synchronous waves were useful for small vessels, and the wave dissipation could be used to

highlight the potentially dangerous seas in spite of its incomplete physics. Further, they argued that statistical predictions of dangerous waves are possible; however, the level of risk is highly dependent on the vessel size, stability properties, ship heading, and speed. From the ship accident statistics, they concluded that both the moderate, but rapid developing seas, as well as the seas more serve than the averaged local wave climate are closely related to the higher risk of ship accidents.

According to (Zhang and Li, 2017), the complex sea states with the co-occurrence of wind sea and swell conditions represent

threats to sailing vessels, especially when these conditions include similar wave periods and oblique wave directions. (Zhang and Li, 2017) made an analysis of 58 cases of swell-related ship accidents from 2001 to 2010 using the three-wave parameters such as the significant wave height, mean wave period and mean wave direction They have also done two case studies showing





an agreement with the general conditions of a possible occurrence of dangerous waves based on the statistical analysis. The relationship between rough seas and ship navigation situations was not mentioned due to the lack of necessary ship information.

Statistical analysis of sea states and related ship accidents from the database can give notification of the most dangerous sea parameters on different ship accidents. However, due to the lack of detailed ship motion and dimension information on each ship accident, they underestimated ocean states because of the coarse resolution of the meteorological databases they used and the possible underestimation of wave steepness resulting from the lack of consideration of wave-current interaction; their conclusions may need further improvement using validation with on-board observations and higher-resolution information of

more accurate ocean states, in spite of their unneglectable contribution to provide some worthy information and suggestions to the ship operators.

Besides poor weather and dangerous sea states leading to ship losses or accidents such as hull breaking, grounding, and capsizing; the unexpected rough seas are also an important factor leading to negative influences on ship safety and navigational economics. For example, waves can affect the structural integrity and ship stability through the direct action of waves on ship

hull; waves can also produce an indirect influence on ship stability such as the water on deck and rolling, and the improper crew operations can also increase the risk of ship accidents due to rolling caused by waves. Besides, the added resistance due to waves can increase the engine burden and speed loss, bringing a high risk of engine failure and then the out-of-control movement. Additionally, a large ship motion amplitude such as pitch, roll, deck wetness, slamming, and propeller racing as well as other coupling phenomena caused by ocean waves are also resulting in unexpected influences on ship safety and

navigational economics.

As a continuation of previous studies focusing on the wave modelling of high wind seas in typhoon periods (Chen C. et al., 2013; Chen C., et al., 2015) and the study on wave-current interactions accounting for the Kuroshio current in the East China Sea (Chen C., et al., 2018), the present study further analyses the relationship between ocean states and ship navigational responses. This study employs a combination of high-resolution numerical wave modelling and detailed on-board observations

of ship motion information, which is affected by rough weather conditions. The study is aimed at providing ship operators practical suggestions for safe navigation by identifying the high risk ocean states and also provides a few possible warning criteria, especially for rough-sea navigation.

For this study, a six-year on-board observation of weather, ocean, and ship motion was performed since 2010, using a 20,000 DWT class bulk carrier covering high-risk shipping regions in the Southern Hemisphere. Two rough sea navigation cases have



been studied by (Sasa et al., 2015), and they demonstrated the reliability of our observation system by validating ocean waves via simulations.

Additionally, four rough sea cases were studied for 2 years by using detailed ship motion information collected using on-board recording facilities, with sufficient temporal and spatial variability, covering high-risk shipping regions in the Southern Hemisphere. High spatial and temporal resolution states of ocean surface wind and waves have also been generated using the

wave model WW3. The validation of these two types of data enable us to have further study the effects of ocean states on the safety as well as the economics of ship navigation.

The paper is organized as follows. In section 2, the on-board observation facility and brief information on data collection are presented. The numerical wave simulations are described in section 3. The validation of the model results with observations are presented in section 4. The relationship between ocean states and ship navigation safety is described in section 5. The

conclusions of this study and suggestions for future studies are summarized in section 6.

## 2. On-board observations

A 28,000 DWT class bulk carrier transporting cargos as a tamper has been used for on-board observation, and data were collected from 2010–2016 for this study. The length of the ship (LPP) is 160.4 m and width is 27.2 m, with a draft of 9.82 m when fully loaded, as shown in Table 1. The ship's routes mainly cover oceans between Asian countries and areas of the

Southern Hemisphere such as Oceania and South America, where ocean states are generally rougher than those in the Northern Hemisphere owing to strong swells. Due to the low operational speed of approximately 12 knots, it has limited feasibility to avoid rough ocean states, which negatively influences its safety and efficiency.

**Table 1: Dimensions and propulsion of the bulk carrier for on-board observations**

| | |
|---|---|
| *Length between Perpendiculars (LPP)* | 160.4 m |
| *Breadth* | 27.2 m |
| *Draft* | 9.82 m (fully loaded) |
| *Displacement* | 28,280 t |
| *Block Coefficient* | 0.77 |
| *Prismatic Coefficient* | 0.78 |
| *Operational Speed* | 12.0 Knots |





To understand the relationship between sea state parameters and rough sea navigation more, we have selected 4 observation
cases from all the experiment period for analysis. The crew almost lost control while recording in the logbook (Case 1).
Information on navigation time, ship location and ship loading condition in each case is shown in Table 2.

**Table.2. Ship location, loading, and timing of each observation case**

| Case NO. | Case1 | Case2 | Case3 | Case4 |
|---|---|---|---|---|
| *Time* | Sep, 2013 | Mar, 2016 | Jun, 2013 | Jun, 2013 |
| *Geographical Location & Model domain* | Tasman Sea; 135~180E, 20~60S | South of Australia; 100~160E, 20~60S | South of South Africa; 5~55E, 10~60S | East of South America; 275~335E, 105 20~60S |
| *Loading Condition; Draft (m)* | **Ballast;** 4.08/5.15 | **Ballast;** 5.12/6.24 | **Half loaded;** 8.16 | **Half loaded;** 8.16 |

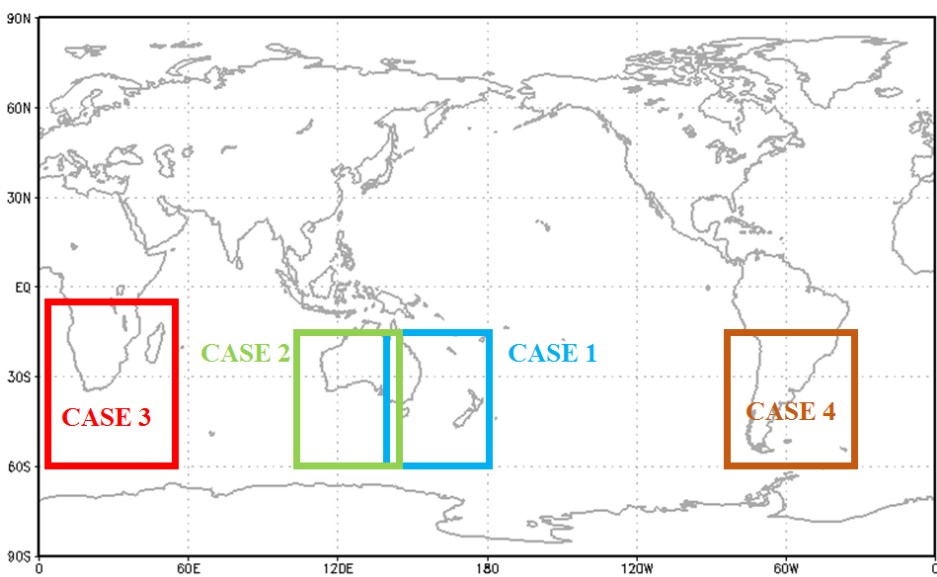

**Figure 1: Geographical locations of all ocean regions set for wave model simulation.**





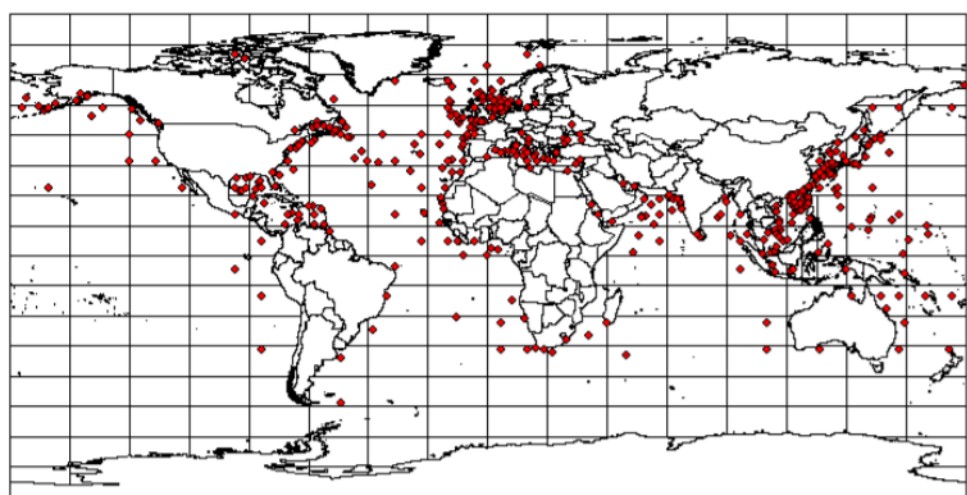

**Figure 2: Worldwide ship accidents (1995-1999) due to severe weather. Source: Lloyd's Marine Information Service (LMIS) casualty database. (Toffoli et al., 2003.)**

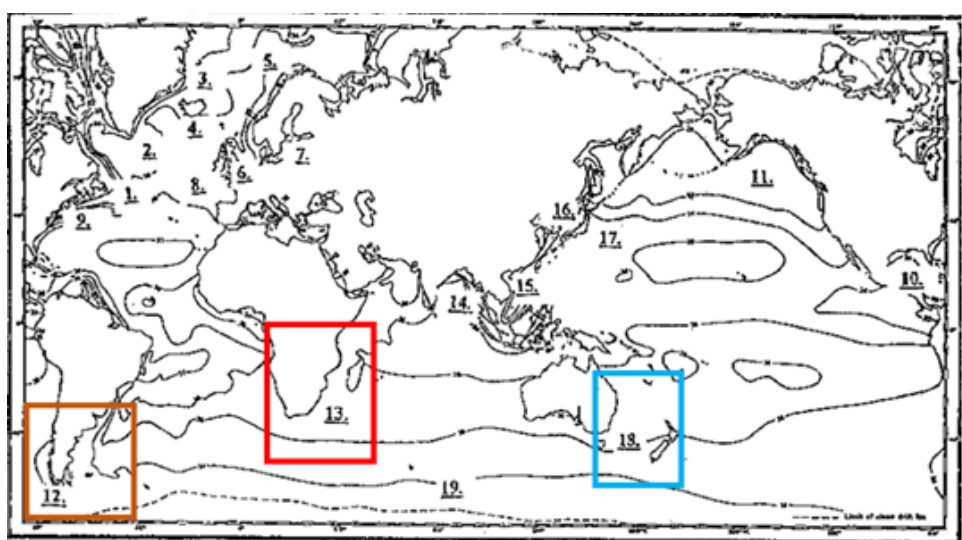

**Figure 3: Freak waves and ship accidents, where 21 specific areas are identified where freak waves have been observed and related ship accidents have been reported. (Kjeldsen, 2004)**



The coloured rectangle in Fig. 1 indicates all ocean regions set for high-resolution wave model simulation for these 4 cases. In Fig. 2, (Toffoli et al., 2003) proposed a worldwide ship accidents (1995–1999) map due to severe weather, using Lloyd's Marine Information Service (LMIS) casualty database. Peter Kjeldsen (Kjeldsen, 2004) identified 21 ocean areas where
unconventional waves occur, causing ship accidents, as shown in Fig. 3.

A comparison of these three figures illustrates the high navigational risk of the selected 6 rough sea navigation cases, especially by the number 12, 13, and 18 provided in Fig. 3, which show a higher risk of experiencing unconventional waves. In addition, according to (Cattrell et al., 2018), the frequency of occurrence of unconventional waves and their generating mechanism is not spatially uniform, and each location is likely to exhibit unique sensitivities. Thus, unconventional waves occurring in ocean
regions of different characteristics are a result of different reasons such as the opposite current effects in the Kuroshio region, the swell and wind sea interactions in the Southern Hemisphere, and the severe storms in the tropical regions. Therefore, detailed information of the on-board observation and the high-resolution wave model simulation are used here for a clear investigation.

The measurements of these observation cases mainly include navigational parameters such as weather data (wind speed, wind
direction, etc), engine data (engine revolution, engine power, and fuel oil consumption as well as the exhaust gas temperatures on all six cylinders of the main engine), voyage data (ship speed, ship position, ship course, loading condition, etc), and the ship's motion (pitch, roll, and yaw). Additional detailed information on this observation instrument and the measurement parameters have been provided by (Sasa et al., 2015). To obtain detailed ship information for an accurate analysis, the ship motion was treated as significant values generated using the zero-up cross method for a 10-min time series with an interval of
0.1 s, and all other parameters are results of the averaged values for a 10-min time series obtained every 1 s.

### 3.    Numerical simulations
### 3.1.    Model descriptions

As a third-generation phase-averaged wave model, the WAVEWATCH III (WW3 model; version 4.18) (Booij and Holthuijsen,
1987; Tolman, 1989; Tolman, 2014) has been used for hind cast wave simulations of all the above-mentioned 4 cases. With an implicit assumption of random phase spectral action density balance equation as those properties of medium (water depth and current) as well as the wave field itself vary on time and space scales that are much larger than the variation scales of a single wave, the WW3 model can solve wavenumber-direction spectra. By explicitly parameterizing all physical processes, such as wind input growing actions, nonlinear resonant wave-wave interactions, wave-bottom interaction, and whitecap
dissipation,  the spectra action balance Equation can be solved as follows (Tolman, 2014):



$$\frac{\partial N}{\partial t} + \nabla_x \cdot (c_g + U)N + \frac{\partial}{\partial k}\hat{k}N + \frac{\partial}{\partial \theta}\hat{\theta}N = \frac{S}{\sigma} \tag{1}$$

$$\hat{k} = -\frac{\partial \sigma}{\partial d}\frac{\partial d}{\partial s} - k \cdot \frac{\partial U}{\partial s} \tag{2}$$

$$\hat{\theta} = -\frac{1}{k}\left(\frac{\partial \sigma}{\partial d}\frac{\partial d}{\partial m} + k \cdot \frac{\partial U}{\partial m}\right) \tag{3}$$


where N is the vector wavenumber spectrum, $c_g$ is the wave group velocity, $U$ is the current velocity, $s$ is the coordinate in the direction of θ, $m$ is the coordinate perpendicular to $s$,and $S$ is the net source term for the spectrum, $\sigma$ is the intrinsic wave radian frequency. In this study, S was determined as the summation of the linear input ($S_{ln}$) to provide more realistic initial wave growth for the consistent spin-up of a model from quiescent conditions (Cavaleri and Malanotte-Rizzoli, 1981); wind

input ($S_{in}$) and wave dissipation ($S_{ds}$) (Tolman and Chalikov, 1996) calculated by a non-dimensional wind-wave interaction parameter; nonlinear wave-wave interaction ($S_{nl}$) using the discrete interaction approximation (DIA) (Hasselmann et al., 1985); and wave-bottom interaction ($S_{bot}$) (Hasselmann et al., 1973) with the empirical, linear JONSWAP parameterization for additional processes in shallow water areas. High-order conservative numerical schemes are used for spatial discretization, and a Courant-Friedrichs-Lewy (CFL) condition exists, binding the discretizations in time and in space.


### 3.2.    Model settings and input data

A two-way nesting method (ww3-multi) with the horizontal resolution of 0.5° and 0.1° for a large (global) domain and the inner-nested (local) domain was used. Considering the CFL condition, the minimum wave propagation time step was set as

330 s and 300 s for the larger and inner domain. For all rough sea cases, the wave model has a spectral resolution of 10° covering 36 directions. Calculated wave frequencies were set from 0.0345 Hz, with a logarithmic frequency factor of 1.1 for 38 steps. Additionally, a one-month spin-up before the time period of ship motion analysis has been run for all cases to start the model from a resting condition.

To simulate an ocean wave model accurately, a critical input is the "forcing" by wind fields: a time-varying map of wind speed and directions. Meanwhile, the most common sources of errors in wave model results are the errors in the wind field. Therefore, to reduce the model uncertainty originating from the sensitivity of the wind input, as illustrated by the studies such as Tolman et al. (2002), Feng et al. (2006), and Campos and Guedes Soares (2016), GPV databases (including both the National Centers for the Environmental Prediction Final (NCEP-FNL), Operation Model Global Tropospheric Analyses

(NCEP/NWS/NOAA/U.S. Department of Commerce, 2000) and the European Center for Medium-range Weather Forecasts Interim Reanalysis (ERA-Interim) (Dee et al., 2011)), have been utilized as the input wind forces for the generations of ocean waves. The NCEP-FNL analyses offer a 1° × 1° grid covering the global region, whereas ERA-Interim offers a spatial





resolution of approximately 80 km (0.75°). Both these wind input sources are updated every 6 h. A linear interpolation method was utilized for applying these two GPV databases (1° and 0.75°) to wave modelling (0.5° and 0.1°).


To calculate the wave-current interaction, the ocean current data Ocean Surface Current Analysis Real-time (OSCAR), which is generated by Earth Space Research (ESR), has been used for wave modelling (Bonjean and Lagerloef, 2002). The ocean current data on a 1/3 degree grid has also been interpolated into 0.1°. Furthermore, the wave-ice interactions are also calculated using the wave model WW3. Thus, sea ice and icebergs have been included in the calculation. For these 4 cases, the sea ice

coverage was calculated in the global domain, using the Nimbus-7Scanning Multichannel Microwave Radiometer, the Defense Meteorological Satellite Program's Special Sensor Microwave Imager, and Special Sensor Microwave Imager Sounder (Cavalieri et al., 1996).

**4.      Model results and validations**

**4.1.      Distributions of ocean surface wind and waves by model simulations**

The ocean waves were calculated using the WW3 model, and the detailed settings are provided in section 3.2. The surface wave distributions at the moment of the maximum pitch motion of all rough sea cases are presented in Fig. 4. Low pressures and swells resulted in strong head, bow, or beam waves during rough sea ship navigation.


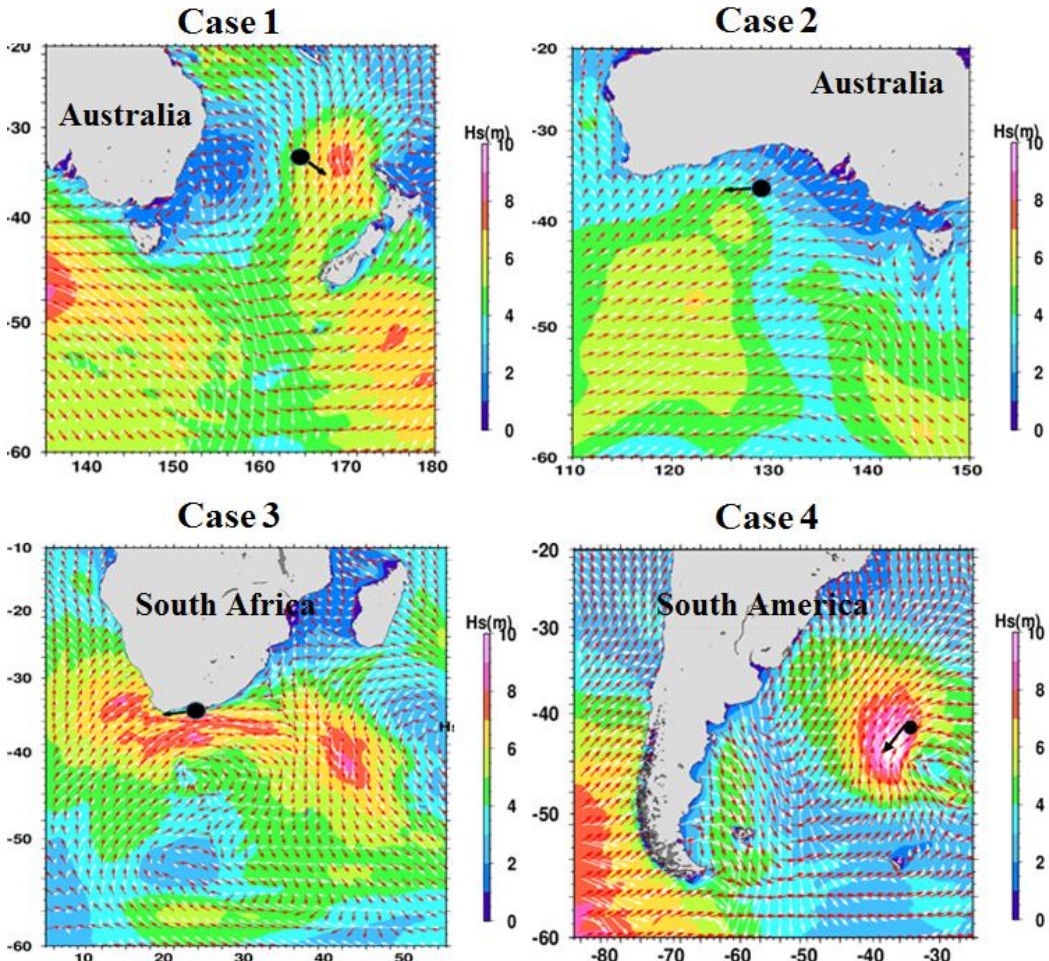

**Figure 4: Ocean surface wave distributions by the WW3 model of all rough sea cases. The shaded areas are significant wave height, white arrows denote wind vector, red arrows represent mean wave direction, black dot denotes the ship position, and black arrows denote ship heading.**


## 4.2.    Validations of simulation results with ship on-board observations

As mentioned at the end of Section 1, observed wind speed of all cases has been averaged for a 10-min time series obtained every 1 s for validation with WRF results, as shown in Fig. 5. Generally, the GPV wind data reasonably replicated the temporal variation of both wind direction and speed, while the NCEP tends to perform better for peak wind than the ERA-Interim.



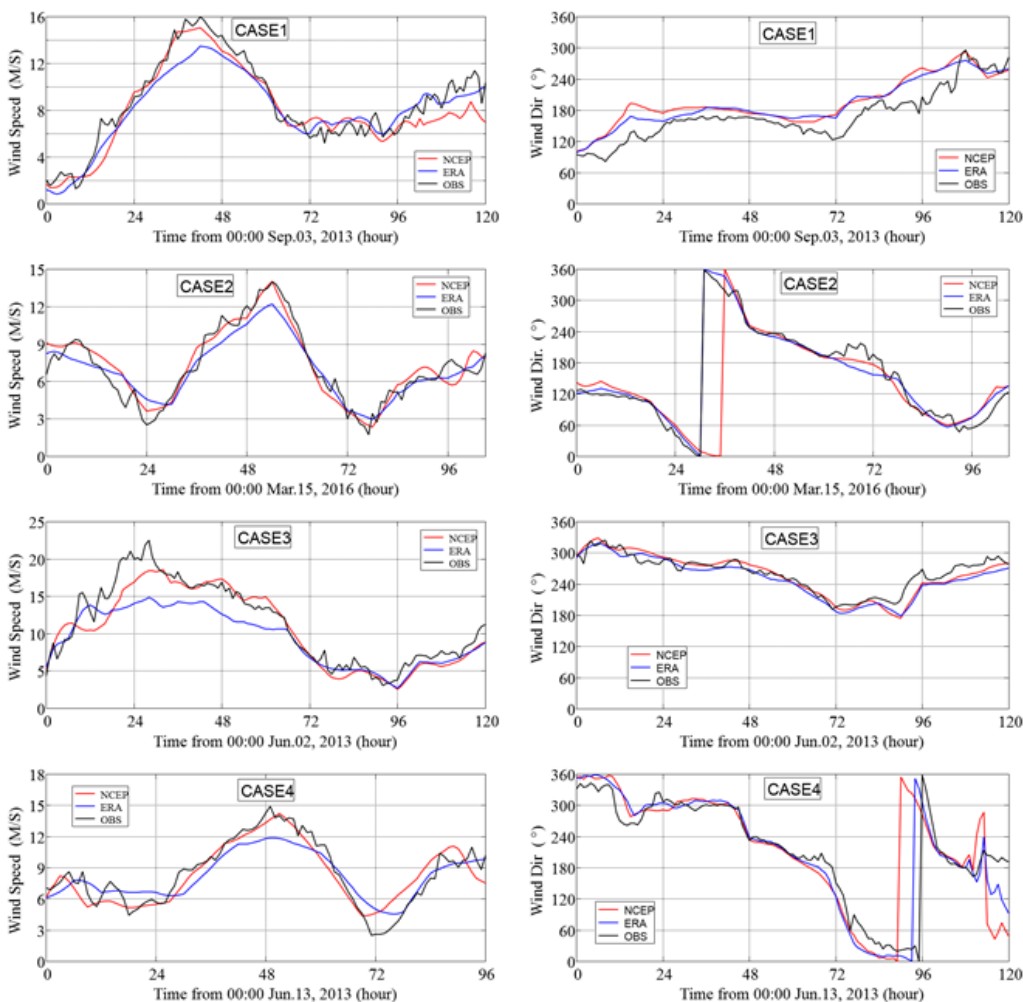

**Figure 5: Validation of on-board observed wind speed with WRF results.**


## 5.        Relationship between ocean state parameters and ship responses

### 5.1.        Ocean states

The correlation among the significant wave height, which is representative of sea severity, and other wave parameters

generated by the WW3 model of all these rough seas is focused on as the first stage, as shown in Fig. 6. The formulas used in

the wave model for generating those wave parameters are also presented in Table 3. Here, the wave energy is E $=$

$\int_0^{2\pi} \int_0^\infty F(f_r, \theta)\, df_r d\theta$, where $\sigma = 2\pi f_r$ is the intrinsic wave radian frequency, and $F(f_r, \theta)$ is the frequency-direction

spectrum. Additionally, a $= \int_0^{2\pi} \int_0^\infty \cos(\theta)\, F(\sigma, \theta)\, d\sigma d\theta$, and b $= \int_0^{2\pi} \int_0^\infty \sin(\theta)\, F(\sigma, \theta)\, d\sigma d\theta$.





**Table 3: Formulas used for generating wave parameters**

| Wave parameters | Formula |
|---|---|
| Hs | $H_s = 4\sqrt{E}$ |
| Mean wave direction | $\theta_m = \mathrm{atan}\left(\dfrac{b}{a}\right)$ |
| Mean directional spread | $\sigma_\theta = \left[ 2\left\{ 1 - \left(\dfrac{a^2 + b^2}{E^2}\right)^{1/2} \right\} \right]^{1/2}$ |
| Mean wavelength | $L_m = 2\pi \overline{k^{-1}}$ |
| Mean wave period | $T_{m02} = 2\pi \Big/ \sqrt{\overline{\sigma^2}}$ |
| Wave steepness | $2\pi H_s \Big/ gT_{m02}{}^2$ |

To improve the ship safety in rough seas and the efficacy of optimum ship routing in reality, different loading conditions should be included. Here, three different groups are divided as the "Half-Loaded" (Case 3, 4), "Ballast" (Case 1, 2), and "Total" (Case 1, 2, 3, 4), which represent the cases of Half-Loaded conditions, Ballast conditions, and the whole cases, respectively. The numbers inside each panel indicate the correlation coefficients of the relevant two-wave parameters. A 10-min average of a time duration approximately 30 days results in a total of 3000 data for these cases, covering high-risk shipping regions in the Southern Hemisphere, as shown in Fig. 1 and Table 2.

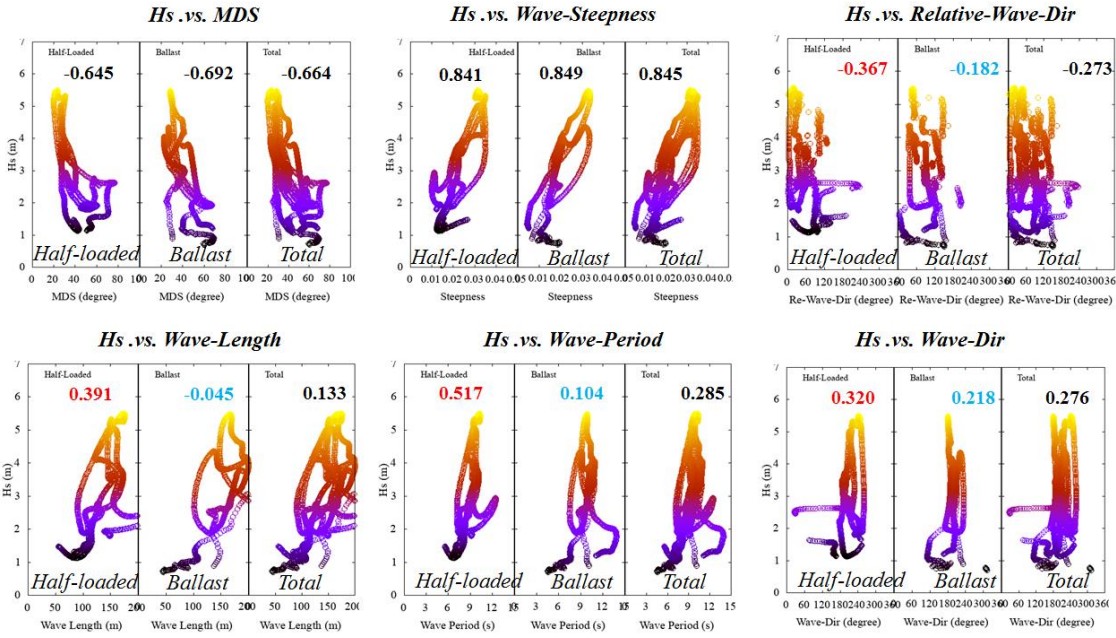





**Figure 6: Correlation among selected wave parameters of all rough sea cases. The "Half-Loaded", "Ballast" and "Total" represent**
**the cases of Half-Loaded conditions, Ballast conditions, and the whole cases, respectively; while those numbers inside each panel**
**are the correlation coefficients of relevant two-wave parameters.**

As shown in Fig. 6, regardless of the ocean states encountered by the two different loading conditions, there are small
differences of MDS and wave steepness between the half-loaded and ballast cases. For instance, the wave height has a strong
positive correlation coefficient with a wave steepness of 0.85, and the mean directional spread is found to decrease with an
enhancement of the significant wave height, with a negative correlation coefficient of 0.66.

However, a stronger correlation exists in the half-loaded cases than that in the ballast cases, when considering other ocean
parameters such as the wavelength, wave period, and wave direction, although the correlation is relatively weak for these
wave parameters. This can be attributed to the larger effects from ocean waves, such as the resonances induced by similar
length and period of ship and encountered waves, are needed for generating ship responses of the half-loaded cases to the
same extent with those of the ballast ones. Details of these ocean states and corresponding ship responses are provided in
section 5.3.

**5.2.      Ship responses**

In addition to those wave parameters, the correlation among observed ship navigation and motion parameters of all rough sea
cases are also considered, as presented in Fig. 7. A relatively strong positive correlation can be found between the pitch and
roll motion (0.660), ship speed and engine RPM (0.760), whereas a strong negative correlation is found between the pitch
motion and ship speed (-0.854).

As shown in the top-left panel in Fig. 7, a stronger correlation between roll and pitch motion can be found in the ballast
(0.838) than that in the half-loaded cases (0.510), indicating a larger influence of ocean waves on the ship motion in ballast
conditions.

As observed in the top-middle panel, as the pitch amplitude increases, the ship operators tend to further reduce the engine
RPM (a higher correlation coefficient of -0.717), but later (when the pitch motion reaches approximately 3 degree in the
half-loaded cases than in the ballast ones (-0.513 and less than 2 degree).

As for the correlation between pitch motion and ship speed, as in the top-right panel, it is observed that the ship operators
preferred to maintain the ship speed to a similar level in both loading conditions by reducing the engine RPM (as mentioned
above) to balance the wave effects and ship motion.

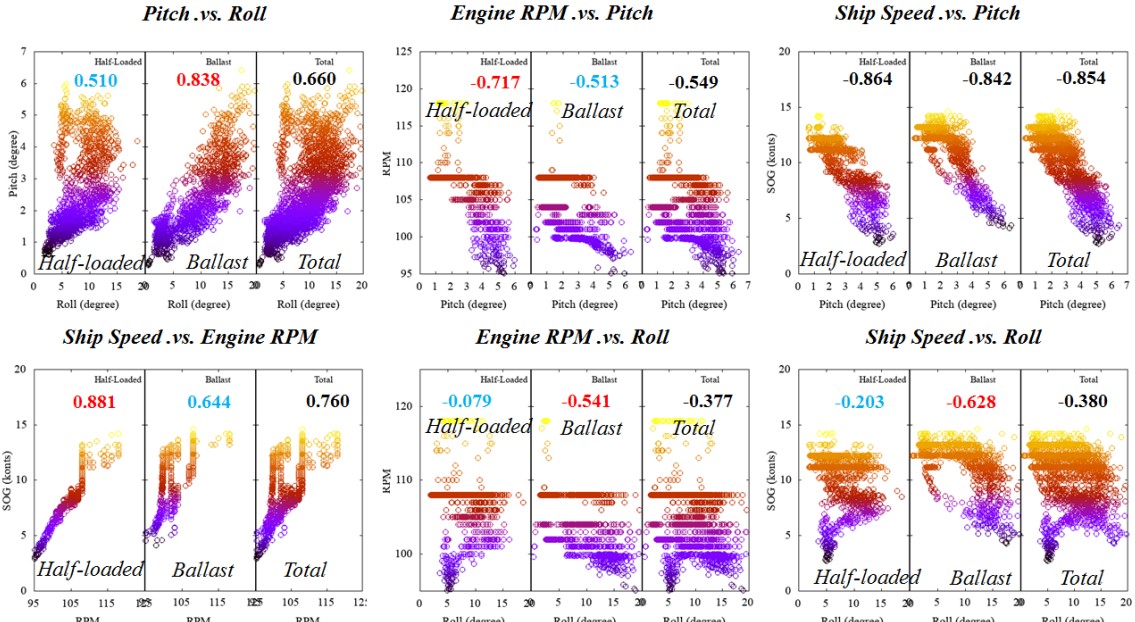

**Figure 7: Correlation between the observed ship navigation and motion parameters of all rough sea cases.**


The bottom-left panel shows that the engine RPM and ship speed loss are more highly correlated in the half-loaded cases than in the ballast ones, indicating a harder control of the ship speed in the ballast conditions due to the more complex coupled pitch-roll motions induced by ocean waves, while the pitch motion occupies a relatively large part of the ship motion in the half-loaded cases.


As observed from the bottom-middle and bottom-right panels, the correlations between the roll motions with other ship responses in the ballast loading cases have a stronger relationship than those in the half-loaded ones, owing to the higher centre of gravity in the ballast loading conditions. For instance, a correlation difference in the relationship between roll motion and engine RPM as well as ship speed for ballast and half-loaded cases are found as large as around 0.46 and 0.43,

respectively.

Lastly, for the half-loaded cases, a pitch motion approximately 3 degrees or a roll motion of approximately 5 degrees can bring a significant drop in engine RPM (less than 100) and SOG (less than 5 knots), as shown in the top-middle, top-right, bottom-middle and bottom-right panels "Engine RPM vs. Roll" and "Ship Speed vs. Roll"; the reason for this can be found

in the top-left panel "Pitch vs. Roll" that the pitch motion is large when the ship suffered a 5-degree roll motion, where the possible pitch-induced ship motion in the longitude direction such as slamming, deck wetness, and propeller racing may lead to a manual operation on the engine and then the voluntary speed loss. For the ballast cases, a pitch motion around 5 degrees or a roll motion around 20 degrees can lead to a drop of engine RPM and SOG.




**5.3.    Ocean states and ship responses**

According to the correlation between these wave parameters and ship responses of all rough sea cases, as shown in Table. 4, both the wavelength and mean wave period have a weak correlation with the ship responses. Therefore, the other four-wave parameters including the Hs, RWD, MDS and the wave steepness will be focused on in the following analysis. Compared

with the roll motion and engine RPM, the wave states tend to have a stronger correlation with the pitch motion and ship speed.

**Table 4: Correlation coefficients of wave states and ship responses of all cases.**

|  | *Pitch* | *Roll* | *Engine RPM* | *Ship Speed* |
|---|---|---|---|---|
| *Hs* | **0.88** | **0.75** | **-0.54** | **-0.67** |
| *Wave steepness* | **0.84** | **0.60** | **-0.59** | **-0.80** |
| *Relative wave direction* | -0.47 | -0.19 | 0.42 | **0.56** |
| *Mean directional spread* | **-0.64** | **-0.54** | 0.43 | **0.51** |
| *Wave length* | -0.01 | 0.04 | 0.02 | 0.20 |
| *Mean wave period* | 0.11 | 0.17 | -0.08 | 0.11 |

**5.3.1.    Significant wave height**

The significant wave height is representative of sea severity. During rough sea navigations, relatively high values significantly influence ship motion and operation. As shown in Fig. 8, the pitch and roll motions increase with the enhancement of significant wave height, with positive correlation coefficients of approximately 0.88 and 0.75.


An increase in significant wave height can result in a drop in the engine RPM and SOG, with negative correlation coefficients of approximately 0.54 and 0.68. As shown in the top-right panel, a correlation difference in the relationship between the significant wave height and roll motion for ballast and half-loaded cases is found to be approximately 0.20, thereby indicating a larger influence of the ocean waves on the roll motion in those ballast cases.

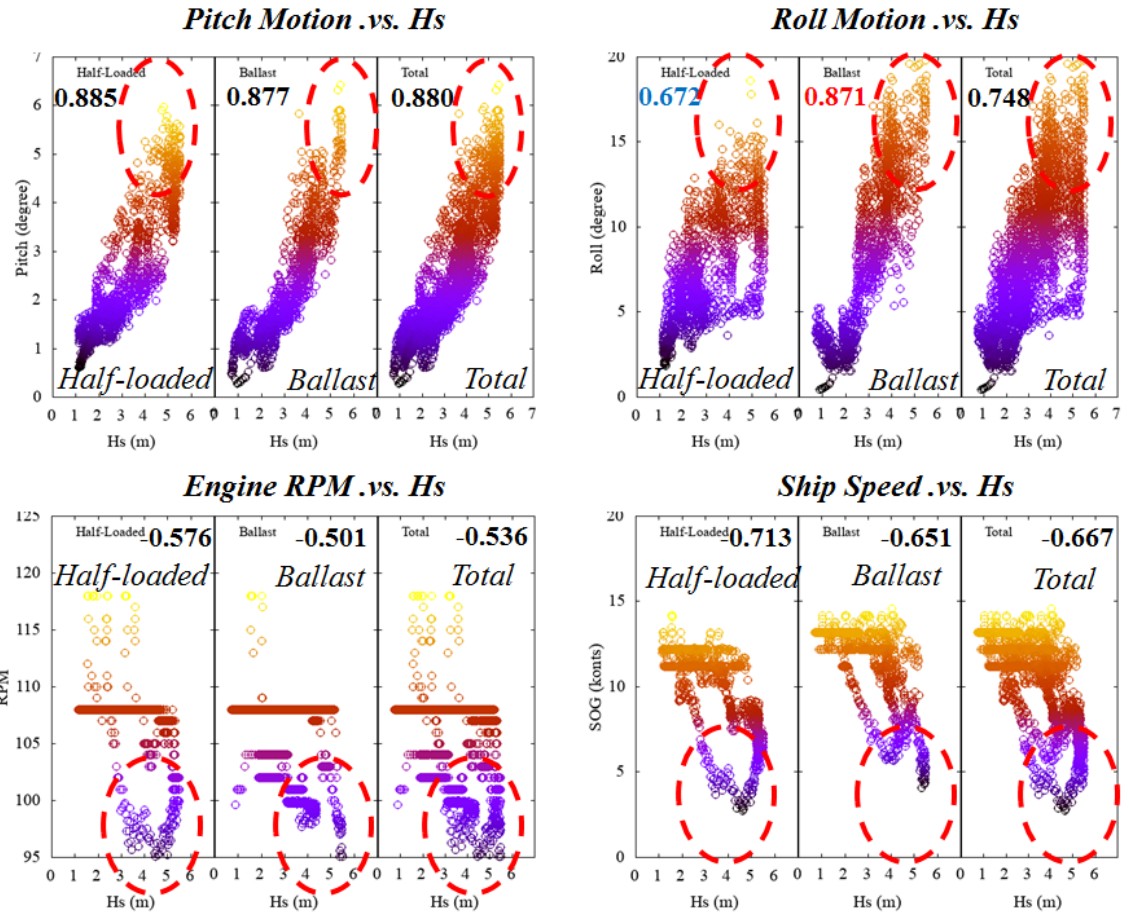


**Figure 8: Correlation between the significant wave height and observed ship navigation and motion parameters of all rough sea cases.**

### 5.3.2. Relative wave direction


The impact of waves on a ship strongly depends on the relative wave direction, as defined in Fig. 9. As an example, the head and bow waves can reduce ship speed by inducing large pitch motion while the beam and quartering waves can affect ship stability and then the reduction of engine RPM and ship speed. Therefore, ships usually navigate perpendicular to the crests in rough seas, with very low forward speed.



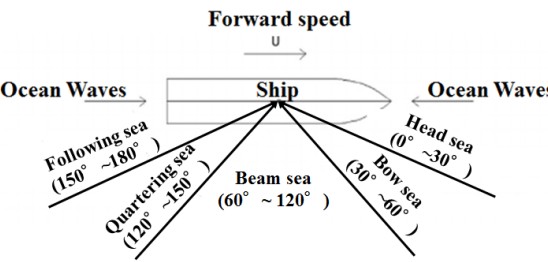


**Figure 9: Relative wave direction defined in this study.**

As shown in Fig. 10, both the largest amplitude of ship motion and significant drop of engine RPM and SOG occurred when the relative wave direction was less than 60 degrees (including both head and bow waves). The relative wave direction

increases from 0 to 120 degrees; its effects on the pitch motion decrease faster than that for the roll motion. When it comes to the correlation coefficients, a difference of 0.3 can be found from two loading cases in the relationship of roll motion and RWD, showing a larger influence of RWD on the roll motion in ballast cases.

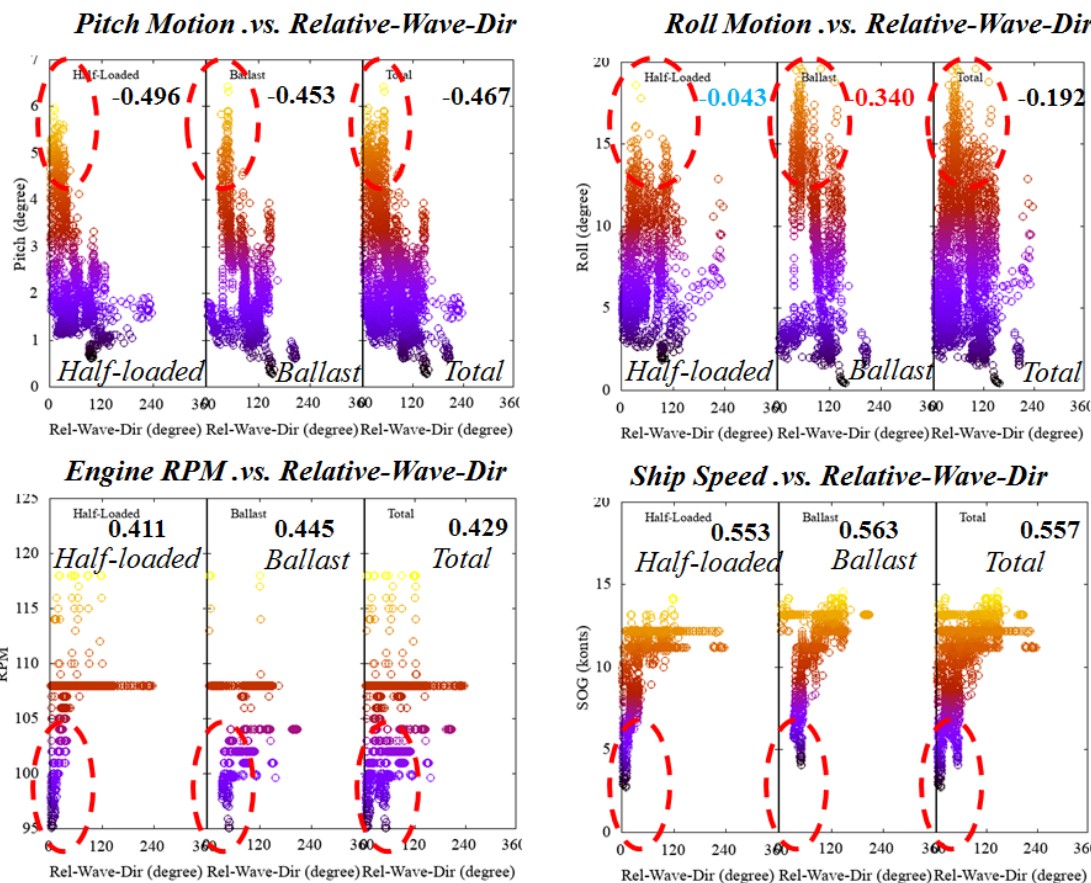




**Figure 10: Correlation between the relative wave direction and observed ship navigation and motion parameters of all rough sea cases.**

It can be seen in the top-left panel that the maximum pitch motion occurred in the head and bow seas in the cases of half-loaded and ballast cases, respectively, and the top-right panel shows that there is another peak of roll motion when the

relative wave direction was around 240 degrees in the half-loaded cases, indicating a quartering wave coming from the stern part. Therefore, head, bow and beam waves brought larger effects in both loading cases. Following quartering waves could also be dangerous because they can induce large roll motion in the half-loaded cases.

The two panels shown below clearly indicate that the head waves exert larger negative effects on the reduction of engine

RPM and speed loss in the half-loaded cases, whereas the bow waves reduced them further in the ballast cases.

### 5.3.3. Wave steepness

As a parameter that points at enhancement of risk of extreme waves, the wave steepness is supposed to have a close

relationship with ship motion (e.g. pitch) and navigation (e.g. speed loss). (Toffoli et al., 2005) pointed out that more than 50% of the incidents took place in sea states characterized by steepness larger than 0.035 (fully developed seas).

As shown in the top-left panel of Fig. 11, a high positive correlation of 0.836 can be found in the relationship between wave steepness and pitch motion, while the wave steepness and roll motion are relatively less-correlated, especially in the half-

loaded cases of 0.523. A correlation difference of 0.25 indicates a larger influence of wave steepness on roll motion in the ballast cases than in the half-loaded cases.

The ship operators started reducing the engine RPM as the wave steepness approached 0.015 in the ballast cases, whereas this reduction was conducted later when the wave steepness was larger than 0.03 in the half-loaded cases, as shown in the

bottom-left panel.

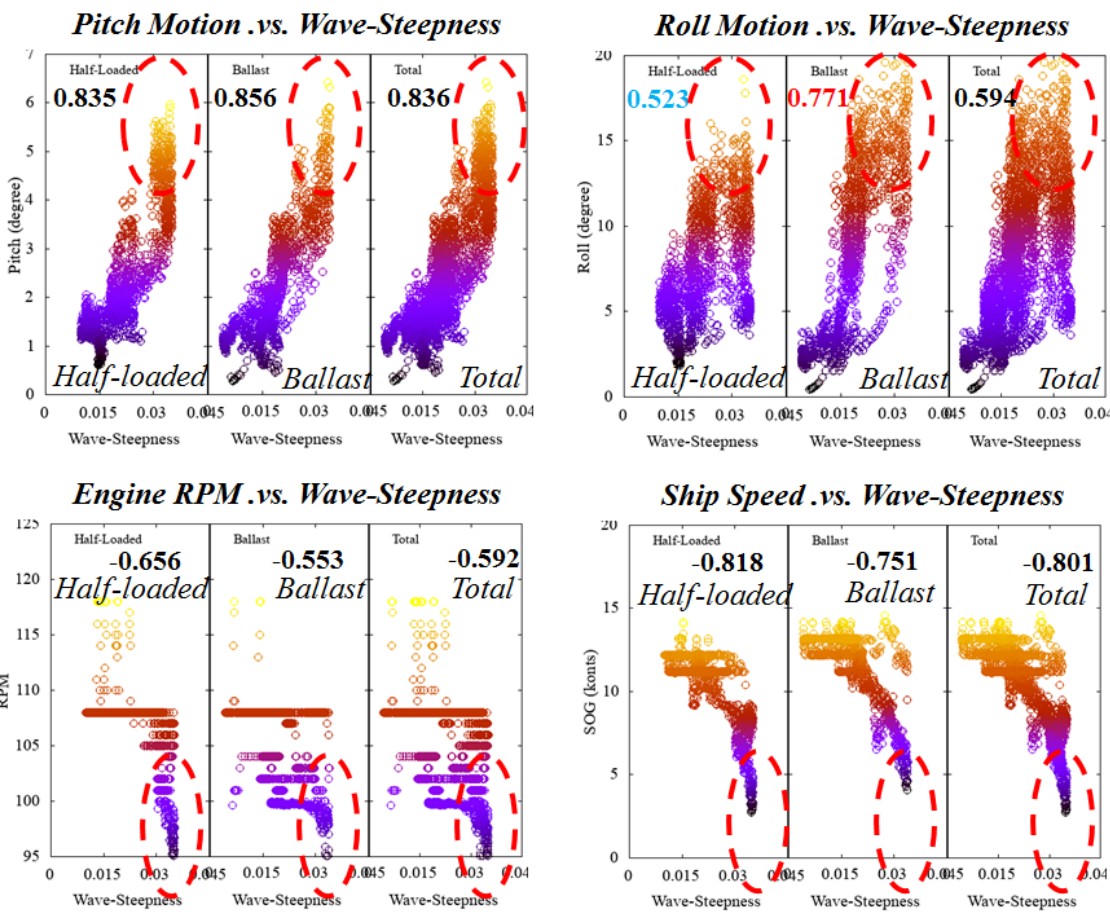

**Figure 11: Correlation between the wave steepness and observed ship navigation and motion parameters of all rough sea cases.**

### 5.3.4. Mean directional spread

According to the experimental tests of two-component directionally spread irregular waves with varying frequencies, directional spreading and component crossing angles made by (Luxmoore et al., 2019) and the reduction of the component directional spreading can increase both the kurtosis and exceedance probabilities. As an important factor to predict the kurtosis and estimate the probability of extreme waves, the mean directional spread is of great importance to identify dangerous ocean regions. Therefore, the mean directional spread was also investigated to better understand the directional information of ocean waves and their effects on ship navigations.

For the whole cases, the pitch and roll motions are negatively correlated with the MDS with coefficients of -0.636 and -0.545, as shown in the top-left and top-right panels in Fig. 12. The MDS had little influence on half-loaded cases (-0.466)




when compared with the ballast cases (-0.685), but the effects of MDS on roll motion reduced more quickly as the MDS increased in the ballast cases, while in the half-loaded cases the MDS kept relatively large effects as the MDS increased and there was a second peak of roll motion when the MDS was around 70 degrees.

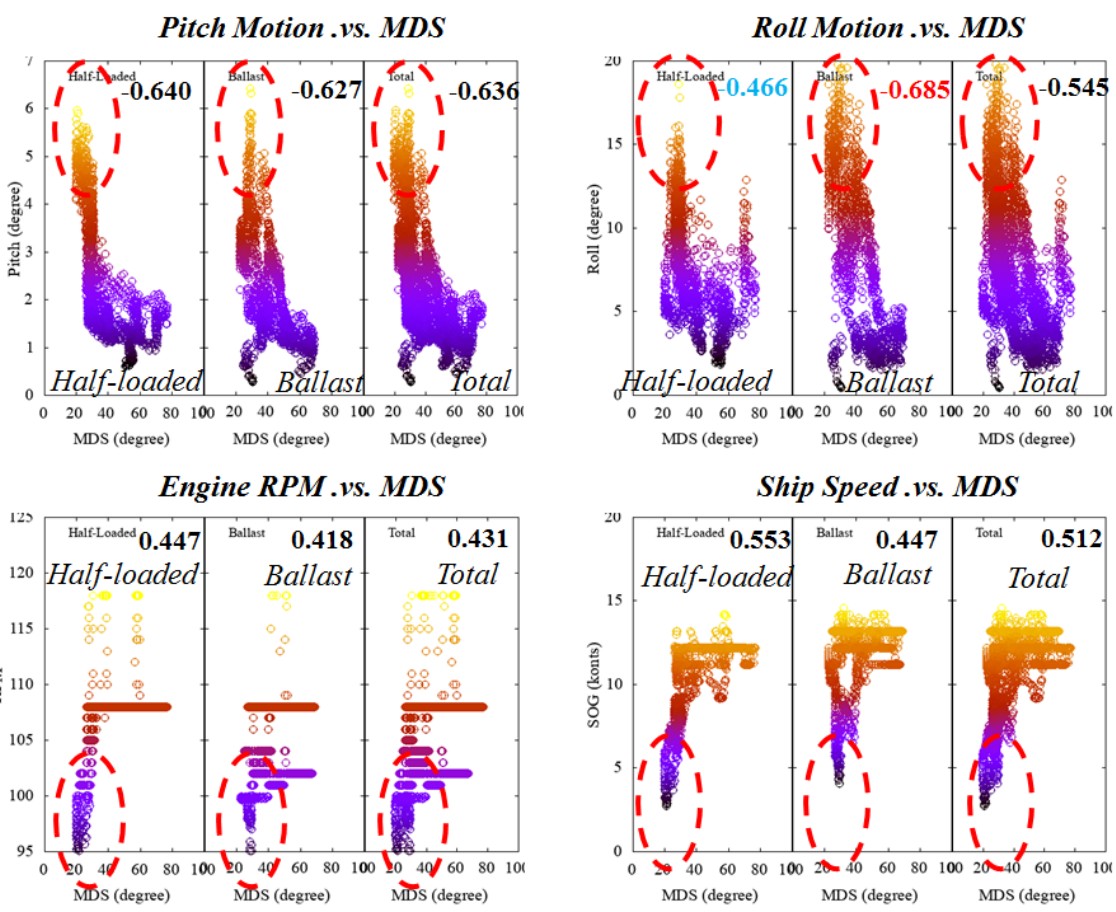


**Figure 12: Correlation between the mean directional spread and observed ship navigation and motion parameters of all rough sea cases.**

### 5.4.    Influences of loading conditions on ship responses to ocean states


Influences of two different loading conditions on ship responses are discussed. As shown in Fig.13, correlation coefficients of ship responses with different loading conditions to ocean states were compared. Ship responses such as the pitch motion (Fig.13-A ), engine RPM (Fig.13-C ) and ship speed (Fig.13-D ) in ballast conditions are of an equal or slightly smaller amplitude than those in the half-loaded ones; large differences exist in the case of roll motion (Fig.13-B ). Owing to a high

GM value and stronger parametric roll resonance in irregular high waves, ocean states including the Hs, MDS, RWD, and




steepness have larger influences on roll motion in the ballast conditions than those in the half-loaded cases, and the differences of correlation coefficients are not less than 0.2.

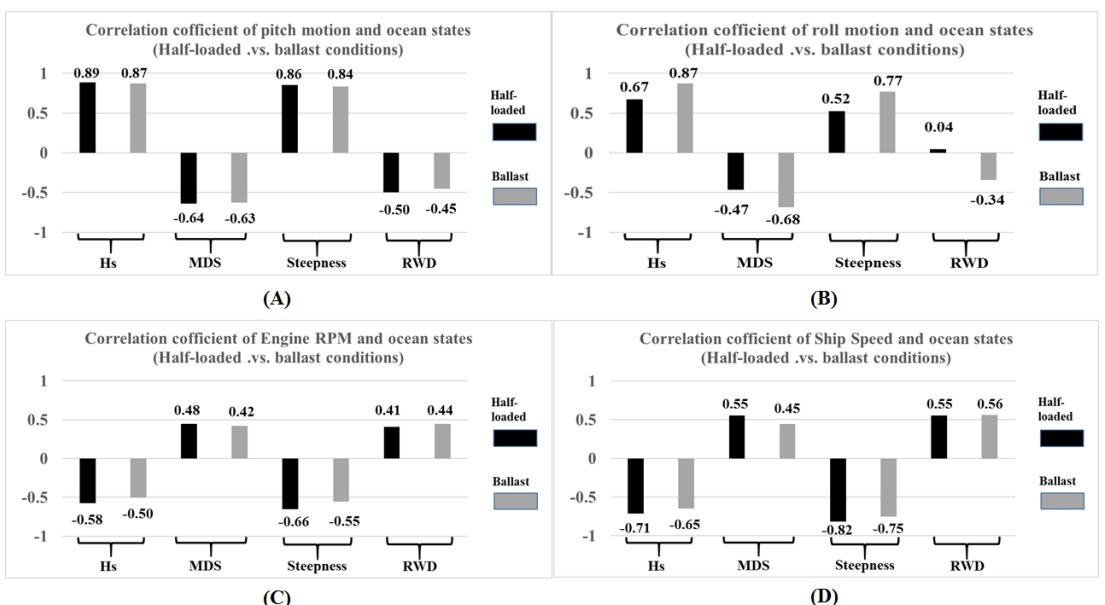

**Figure 13: Differences of ship responses to ocean states in two loading conditions**

Therefore, compared with other ship responses affected by ocean waves in the half-loaded conditions, the roll motion should be paid more attention when the ship is in ballast conditions. As mentioned before, the ship operators usually reduced the engine RPM more but later in the half-loaded cases than they did in the ballast ones.


**5.5.      Identifications of warning criteria in ocean states of different levels**

To provide practical suggestions to the ship operators for their decision-making on identifying and avoiding high-risk ocean states, it is of great importance to figure out the various amplitudes of ship responses induced by rough seas of different
levels. In addition to the correlation between each ocean state and each ship response given above, the ship responses are divided into 3 ranges to find out the ship responses to ocean states of different levels, and the detailed values are given in Table. 5.

**Table. 5. Ranges of ship responses to ocean states**

| | Pitch (degree) | Roll (degree) | Engine RPM | Ship Speed (knots) |
|---|---|---|---|---|
| | | | | |



|  |  |  | (percentage of reduction) | (percentage of reduction) |
|---|---|---|---|---|
| **Large responses** | 4~ | 15~ | ~100 (16.6%) | ~8 (42%) |
| **Modest responses** | 2~4 | 5~15 | 100~105 (7.5% ~ 12%) | 8~10 (24%~42%) |
| **Small responses** | ~2 | ~5 | 105~ (2.7%) | 10~ (24%) |


Identifications of warning criteria in ocean states of different levels are given in Fig. 14, where ship responses to four important ocean parameters are divided into three levels as large, modest and small responses. Ship operators do not need to take measures when the ship is in the blue regions (small responses). They should start to reduce the engine RPM to balance the propulsion and wave resistances or change ship courses in the purple regions (modest responses) to avoid those red

regions (large responses), where they may lead to ship motion of large amplitudes and possible damage to cargoes or ship's hull.

For instance, ocean states with averaged Hs over 5 m, 4 m, and 1.5 m are found for the large, modest and small ship responses in half-loaded cases, while relative smaller waves are found for those ship responses in the ballast cases as 3.5 m,

3.8 m, and 1.3 m. A difference of 1.5 m between two loading conditions can be found for the large ship responses. Ship operators need to reduce ship speed or change course when the ship was experiencing a modest response, with an averaged Hs around 4 m in rough sea cases. Besides, two out of the three ship accidents occurred in ocean regions under 4 m (Toffoli et al., 2005).





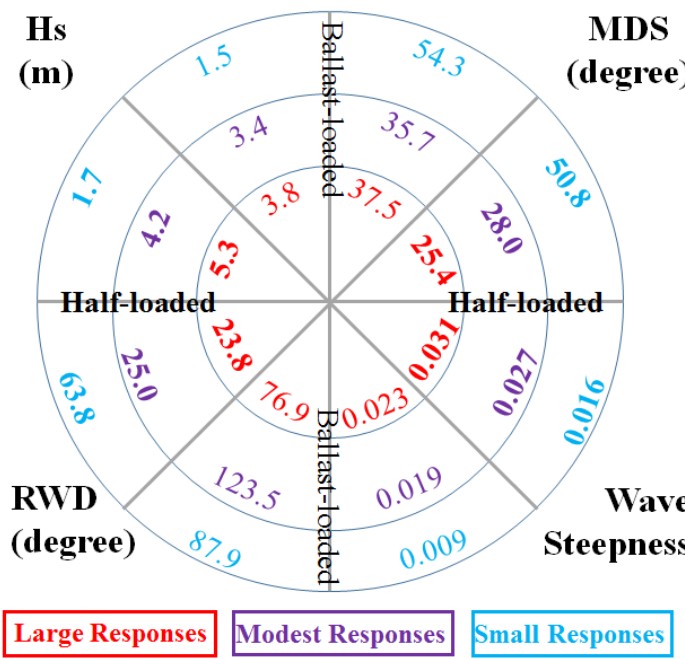

**Large Responses** | **Modest Responses** | **Small Responses**


**Figure 14: Identifications of warning criteria in ocean states of different levels. Ship responses in both half-loaded (in bold font) and ballast cases (in normal font) to the MDS, Hs, RWD and Wave Steepness are shown in the 1st, 2nd, 3rd, and 4th quadrant, respectively. The red, purple and blue colored numbers represent large, modest and small ship responses, respectively.**

When it comes to the averaged RWD for each ship response, a difference of 53 degrees, 100 degrees, and 20 degrees can be found between two loading conditions for the large, modest and small ship responses, respectively. Head seas generate modest and large ship response in half-loaded cases, while beam seas tend to affect more in ballast cases.

For half-loaded cases, large ship responses also occurred in ocean regions with a wave steepness of 0.031 (0.0346 is for fully
developed wave, according to the Pierson-Moskowitz spectrum), while (Toffoli et al., 2005) found that 60% of ship accidents occurred in sea states with wave steepness of 0.03~0.045 from his studies on the relationship of ocean states and ship accidents, shown in Fig. 15. On the other hand, a smaller wave steepness of approximately 0.02 was experienced for large and modest ship responses in ballast cases, indicating a weaker capability to operate the ship over ocean regions with high steepness as that in the half-loaded cases.


Similarly, an MDS difference of around 10 degrees is found between half-loaded and ballast cases for both large and modest ship responses. Also, corresponding with a limited MDS of 25 degrees, the crossing seas are of high probability (98%) for ship accidents (Toffoli et al., 2005), which can also be found in Fig. 14 and Fig. 15 that large and modest ship responses of the half-loaded cases occurred in similar ocean regions. In addition, (Bitner-Gregersen & Toffoli, 2014) also found that the





maximum wave height was affected by crossing angle with a peak around 40 degrees in their numerical studies using

directionally spread crossing seas, while the ship in ballast conditions also experienced modest (35.7 degrees) to large

responses (37.5 degrees) in our study.

**Figure 15: Comparisons of the present study with that of (Toffoli et al., 2005).**

6.    **Conclusions**



Different from model tests focusing on wave effects on ship models with human-designed waves, a detailed analysis of the

relationship between ship responses and ocean states was done in real rough seas with real-time observations in combination

with high-resolution wave hind cast. Information on ship motion, engine operation, ship speed loss, and ship dimension is

available for every moment in these rough seas by on-board data-recording devices. Two widely used GPV datasets

including the NCEP-FNL analyses and ERA-Interim have been used to provide ocean surface wind force to the wave model.

The ocean surface wind was validated using the on-board observations, and the relationship between simulated wave

parameters and observed ship responses were quantitatively analysed and suggestions for ship safety under severe weather

conditions were also given.

At first, the relationship among different wave parameters in actual rough seas was studied using high-resolution wave

models, showing a strong positive correlation of 0.85 between the Hs and wave steepness and a negative correlation of 0.66

between Hs and mean directional spread. Generally, the relationships between Hs and other parameters such as the

wavelength, wave period, wave direction as well as the relative wave direction were not strong, although a stronger

correlation of these relationships has been found in the half-loaded cases than that the ballast-loaded ones.

Secondly, an analysis of the correlation among the observed ship responses shows a strong positive correlation between the

pitch and roll motion (0.66), ship speed and engine RPM (0.76); a strong negative correlation relationship between the pitch

motion and ship speed (-0.85). The ship operators reduced the engine RPM more, but later in the half-loaded cases, than they

did in the ballast ones. Additionally, due to a higher gravity center, the correlation between roll motions with other ship

responses is stronger in the ballast cases than that in the half-loaded cases, and the differences of correlation coefficients are

not less than 0.2.


Thirdly, studies on the ship responses to ocean states show a stronger correlation between ocean states with the pitch motion

and ship speed than those with roll motion and engine RPM, and the wave parameters such as wavelength and mean wave

period have a relatively weak correlation with ship responses. With the enhancement of ocean states such as the Hs and wave

steepness, the amplitude of pitch and roll motion increases while the engine RPM and ship speed decrease. In addition to the

head, bow and beam waves, the following quartering waves could also be dangerous because they could also induce large

roll motion in the half-loaded cases.

Lastly, important ship responses to ocean states of different levels are also given, quantitatively showing the relationship

among ship navigational parameters such as pitch and roll motion, engine RPM and speed loss with those ocean parameters

including the significant wave height, relative wave direction, mean directional spread and wave steepness. For ships of

similar dimensions, these results can provide practical suggestions to ship operators on identifying and avoiding the possible



high-risk ocean regions, thus enabling them to reduce the negative effects on navigational safety and economy induced by unexpected large ship responses, as recorded in the logbook for one case (the first case in the Tasman Sea) that the crew almost lost the control of the ship in the high waves. In addition, such practical suggestions will also help improve the safety

of Autonomous Surface Vehicles operation in harsh ocean environments in future studies.

## 7.    Acknowledgment

We appreciate the National Centers for the Environmental Prediction Final Operation Model Global Tropospheric Analyses

and the European Center for Medium-range Weather Forecasts Interim Reanalysis for providing the data used in this study. We also appreciate the developing group of Weather Research and Forecasting Model and WAVEWATCH III. We appreciate Shoei Kisen Kaisha Ltd., Northstar Shipping Management Ltd., the bulk carrier's crew, and shipping agents in Singapore and Australia for their assistance in conducting this study. This study was financially supported by Grant-in-Aid for Early-Career Scientists '19K15251' (2019–2021, represented by Chen Chen), Scientific Research (B) (2016–2018,

represented by Kenji Sasa) and Fostering Joint International Research (B) (2018–2022, represented by Kenji Sasa) under Grants-in-Aid for Scientific Research, Japan Society for the Promotion of Science. This study was also supported by the Croatian Science Foundation under the project IP-2018-01-3739.

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
