# Peer review of "Identification of early warning criteria for rough sea ship navigation using high-resolution numerical wave simulation and shipboard measurements"

_Natural Hazards and Earth System Sciences, 2019_

## Referee Comment (RC1) · Anonymous Referee #1 · 3 Mar 2020

This paper addresses the risk of operating ships in severe weather conditions and aims at providing decision support for ship navigation. It studies the relationship between weather conditions and motion responses of ships by comparing numerical wave modelling with on-board measurements of ship motions from a bulk carrier. This is an important issue for maritime safety, and also influence optimal weather routing that may lead to more cost-efficient and environmentally friendly shipping. Hence, the paper addresses an important issue for the maritime industry which is, in principle, relevant for publication. However, I feel the quality of the paper is too poor to warrant publication in

a quality international journal. The language and presentation of the research is poor and at times very imprecise, and also the proposed warning criteria seems very simplistic and based on a very simple analysis of correlation between various variables. Hence, I don't find this paper to be interesting enough to recommend publication. On another note, I do not find NHESS to be the most relevant channel for presenting this research and would suggest that the authors possibly submit a revised version to a more ship-focused journal.

Some of my main concerns are given in the following:

- The "warning criteria" are based on a rather simple statistical analysis (essentially only studying correlations between sea state variables and ship motion responses). They seem to be very simplistic and I wonder if not state-of-the art numerical response calculations could be used to obtain better decision support. I guess you may easily estimate the effect of the different sea state variables on ship responses by performing a set of numerical simulations using hydrodynamic models etc (for any loading condition). How does your proposed method compare with such methods and what is the benefit of doing the simple correlation analysis to suggest warning criteria. However, I believe the collected dataset is interesting, and possibly this could be used to validate such numerical models (in addition to model tests, which are commonly used to validate ship model simulations).

- Some ships are equipped with a hull monitoring system to advise about operation in high seas. This is not mentioned in the paper, and a discussion on the effect of such systems, perhaps in combination with other methods for decision support could be relevant in the introduction or in section 2. Such systems measure stresses and accelerations in the ship hull and would provide additional information to what is collected in your case study. You could get ship structural response in addition to ship motion response that may be important in terms of safety and reliability.

- You study the correlation between sea state variables and ship response variables,

and this is a major part of the paper. The times series are obviously highly auto-correlated. It is well known that strong serial correlation may give cross-correlated time-series even for independent variables, so the correlation coefficients given in Fig 6 should be interpreted with care. A note could be included on this (perhaps with reference). Moreover, what insight do you really get from all the correlations you find for the sea-states and ship response variables in sections 5.2/5.3? On p. 13 you state that the ocean waves will have a larger influence on ship motions in ballast conditions. How can you conclude on that from higher correlation between roll and pitch?

- You study the correlation between sea states and ship responses. However, you only study one response/effect at a time. How do you account for interaction effects? That is, the effect of one parameter will be influenced by the value of another. How do you account for possible confounding effects? Could you gain insight if you try to fit a statistical regression model to these data, e.g. explaining the ship responses by the sea state variables. In such models you could include interaction terms to account for such dependencies and could perhaps give more insight than merely studying the (linear, I assume, but you do not say) correlation coefficients. Also, how statistically significant are the correlations you estimate? Particular with respect to section 5.4 and Fig 13 where you compare the correlations for loaded and ballast conditions this is a relevant question.

- When you compare ship responses and what you refer to as navigation – can you assume that different behavior is only due to weather conditions? Is it the same crew operating the ship in all four situations? It is well known that different seafarers may respond differently, for example. Do you have control of other influencing factors (human factors etc.) For example, you state that ship operators usually reduces engine RPM more, but later, in half-loaded cases than in ballast conditions. Can you say this from just two cases of ballast and two cases in loaded conditions? Many other factors than the loading condition could be at play here.

- The warning criteria in Figure 14 is the main results of this study, as I understand it,

and I wonder if the is based on a weak foundation (a crude correlation analysis for four situations). Does this really push state-of-the art in weather warning criteria?

Minor comments:

- The language is at times poor, and thorough language vetting is recommended. The following are merely a few examples from the first parts of the paper but proof-reading by a native speaker is recommended throughout the paper.

o "Due to an increase in ship size and number, shipping activities frequently lead to a higher possibility of ship accidents and increased safety risk to human beings, property losses, and the pollution of ocean environments.". I understand what you try to say, but this is poorly formulated. ..... frequently lead to a higher possibility... What is meant by frequently? And higher than what?

o "Therefore, marine weather information, including an accurate forecast of extreme ocean surface wave states." This sentence has no verb and makes no sense.

o These are just two examples, but the language needs to be improved for this to be published in a quality international journal.

o Check throughout the use of definite/indefinite form (e.g. "the"), and singular/plural forms.

- On p. 3 you mention different failure modes and potential problems related to waves. How about fatigue? I guess operation in severe weather can lead to increased fatigue on ship hull which may ultimately lead to failure. Fatigue is a cumulative effect and perhaps somewhat different from e.g. capsize and grounding, but it seems relevant to include here (possibly, with relevant references).

- Why do you give two drafts for case 1 and 2 in Table 2?

- On p.7 you mention 6 rough sea navigation cases. However, previously you only mention four cases. Check and update.

- Your statement on p.8 should be backed up with a reference: "The most common sources of errors in wave model results are errors in the wind field". There are several other sources of errors. Can you validate this statement by a reference? Alternatively, just say "one of the most common errors..."

- You state that wind input is important but use low-resolution wind input to drive the models. Will linear interpolation (in both space and time, or only in space? Do you assume six-hourly stationary conditions?) give accurate results? Could you use other downscaling methods (physical/statistical) to gain better results? Especially since you argue that the wind forcing is one of the most important sources of error of the numerical wave modelling.

- Abbreviations should be spelled out on first use. For example, what is WRF? (Weather Research and Forecast model??). MDS = Mean directional spreading? RWD = relative wave direction? (relative to wind? Relative to ship heading? OK, this is defined in Fig. 9, but not when it is first mentioned). RPM = revolutions per minute. GM = metacentric height (not obvious to all readers of this journal). Check throughout that all abbreviations are explained.

- What do you mean by "total" or "whole cases" on p. 12? I guess I understand what you mean (that you group statistics from all cases together?) but it should be better explained.

- What do the colors in Figs 6, 7, etc. represent? Time? Value of one of the variables? This should be explained.

- Caption of Figs 6, 7, ... should be revised. You are not showing correlations per se, but scatterplots of selected wave parameters to illustrate correlations. I believe you could be more precise.

- This sentence on p. 13 does not make any sense: "As observed in the top-middle panel, as the pitch amplitude increases, the ship operators tend to further reduce the

engine RPM (a higher correlation coefficient of -0.717), but later (when the pitch motion reaches approximately 3 degree in the half-loaded cases than in the ballast ones (-0.513 and less than 2 degree)." For one, the parentheses do not match. I can understand what you want to say, but you do not say it very well. Also, you continue to say that operators prefer to maintain speed, but the figure clearly shows that speed decreases as pitch increases. What do you try to say here?

- What do the dashed rings in Fig. 8, 9, . . . represent? They are not discussed and should be removed (or discussed).

- On p. 20, the following sentence is very imprecise: "Ship responses such as . . . in ballast conditions are of an equal or slightly smaller amplitude than those in the half-loaded one". You are not comparing ship responses, but correlations between ship responses and sea states. Re-phrase to be more precise.

- On page 22, values of Hs 5m, 4,m and 1.5m are not the same as in Figure 14. Moreover, 3.5m, 3.8m and 1.3m is given in incorrect sequence (large, modest, small).

- The presentation of the warning criteria in Fig. 14 is counter-intuitive, with large responses corresponding to small radius in the circle, and small responses far out with large radius. A minor issue, but strange that you chose to present it this way.

- Reference list must be updated. List all authors not only et al. in the references list. Moreover, several citations in the text cannot be found in the reference list. E.g. Chen et al. 2013; 2015; 2018, . . .

---

## Author Comment (AC1) · 21 Mar 2020

Dear Anonymous Referee #1

On behalf of all Co-Authors, I want to express our appreciation to the Anonymous Referee #1 for his/her valuable comments which we believe is greatly helpful for the improvement of our manuscript. We agree with most of those comments, and we have put our addresses to all of those comments as a supplement PDF file for the confirmation of both the Handling Editor and the Anonymous Referee #1. We hope our

addresses will be satisfying and the revised submission has become qualified to be published in the NHESS to share our research with others. Thank you again and we are looking forward to hearing from you.

Best regards, ChenChen

Please also note the supplement to this comment:
https://www.nat-hazards-earth-syst-sci-discuss.net/nhess-2019-399/nhess-2019-399-AC1-supplement.pdf

**Supplement:**

*Dear Anonymous Referee #1:*

*Thank you very much for your kind time on my submission!*

*I really appreciate your valuable and poignant comments (which I think are the best ones I have ever read, honestly speaking) on helping improving the present manuscript, and we have the following addresses to your comments. We hope our responses and modifications are satisfying. Modifications in the resubmission are given with a yellow-color background here in our addresses.*

*ChenChen*

**Main concerns**
**Comment #1:**

This paper addresses the risk of operating ships in severe weather conditions and aims at providing decision support for ship navigation. It studies the relationship between weather conditions and motion responses of ships by comparing numerical wave modelling with on-board measurements of ship motions from a bulk carrier. This is an important issue for maritime safety, and also influence optimal weather routing that may lead to more cost-efficient and environmentally friendly shipping. Hence, the paper addresses an important issue for the maritime industry which is, in principle, relevant for publication.

However, I feel the quality of the paper is too poor to warrant publication in a quality international journal. The language and presentation of the research is poor and at times very imprecise, and also the proposed warning criteria seems very simplistic and based on a very simple analysis of correlation between various variables. Hence, I don't find this paper to be interesting enough to recommend publication. On another note, I do not find NHESS to be the most relevant channel for presenting this research and would suggest that the authors possibly submit a revised version to a more ship-focused journal.

**Address to comment #1:**

Thank you for your comment!

We appreciate very much your kind words agreeing with the motivation of our study!

We are here dividing this comment into three parts (**A:** language problem; **B:** data analysis; **C:** journal selection) and then making three part of addresses (shown as **A, B** and **C** as follows).

**Address to comment# A (language problem):**

Regarding to the poor language and presentation, as again pointed out in another minor comment "The language is at times poor, and thorough language vetting is recommended. The following are merely a few examples from the first parts of the paper but proof-reading by a native speaker is recommended throughout the paper", we are sorry about it but **we indeed used the English check service from the company "Editage"…** Anyway, we have revised the language in the resubmission. Please check it.

**B: data analysis**

a.  Then, about your comment "*proposed warning criteria seems very simplistic and based on a very simple analysis of correlation between various variables*", we want to address it together with another two related comments:

"*You study the correlation between sea states and ship responses. However, you only study one response/effect at a time. How do you account for interaction effects? That is, the effect of one parameter will be influenced by the value of another. How do you account for possible confounding effects? Could you gain insight if you try to fit a statistical regression model to these data, e.g. explaining the ship responses by the sea state variables. In such models you could include interaction terms to account for such dependencies and could perhaps give more insight than merely studying the (linear, I assume, but you do not say) correlation coefficients. Also, how statistically significant are the correlations you estimate? Particular with respect to section 5.4 and Fig 13 where you compare the correlations for loaded and ballast conditions this is a relevant question.*"

**Address to comment# B-a:**

To improve the paper as well as figure out the interaction terms to account for such dependencies as you said and recommended, a statistical regression model as well as statistically significances of these data has been added to explain the ship responses by the sea state variables in the end of Section.5.

Modifications in the resubmission are as follows:

Additionally, the interaction effects, meaning the effect of one parameter influenced by the value of another one, has also been studied using statistical regression model to further explain the relationship among ship responses and various sea state variables, as shown

in Table. 7. The first four rows are the regression model of wave parameters effects on each ship navigation parameter, and the last two rows focus on the effects of ship motion (pitch and roll) on RPM and ship speed loss.

At first, large "R-squared" values (marked as bold font numbers in Table. 7) of the regression model of "Pitch", "Roll" and "SOG" against all four wave parameters (as shown in the first, second and third row in Table. 7) as well as "SOG" against two ship navigation parameters "Pitch" and "Roll" (as shown in the fifth row in Table. 7) show a high extent of the explanation of these parameters. While other low "R-squared" values (marked as italic font numbers in Table. 7) give a low percent of variance explained. For instance, the fourth and last row giving the regression model of "RPM" against wave and ship motion parameters show a high possibility of influence of other crew operations (such as voluntary speed loss) on Engine RPM rather than the direct effects from ocean waves and wave-induced ship motion.

It can also be found from the regression model of "Roll" and "SOG" in the second, third and last row that spurious correlations, as shown in Fig.14, Fig.15, and Fig.10; exist between "Roll" and "Wave Steepness", "SOG" and "MDS", "RPM" and "Roll", respectively.

**Table. 7. Relationship among ship responses and various sea state variables by statistical regression.**

| Regression model for various ship navigation and wave parameters | Multiple R-squared |
|---|---|
| 1. Pitch=0.3023+0.5877*Hs−0.0032*MDS+32.7620*WS−0.0055*RWD | **0.85** |
| 2. Roll=1.5923+2.3377*Hs−0.0343*MDS+0.0112*RWD | **0.58** |
| 3. SOG=13.6654−0.1427*Hs−175.1856*WS+0.0178*RWD | **0.73** |
| 4. RPM=108.8538−1.0308*Hs+0.0140*MDS−65.7744*WS−0.0075*RWD | *0.26* |
| 5. SOG=14.1316−1.9723*Pitch+0.1713*Roll | **0.76** |
| 6. RPM=109.6030−1.7790*Pitch | *0.30* |

Significant difference has also been added using a statistical method, the calculation of P value between two loading conditions, as shown in Table.5 in the resubmission. Results show that that except for the Hs, all the p values between the two loading conditions are 0, implying a significant difference between them.

Modifications in the resubmission are as follows:

To have a deep look at the difference between above-mentioned two loading conditions the significant test has been done, and the p values are given in Table. 5, where

the "Source" is the parameter for which the significant test has been made, the "SS" is Sum of squares, the "df" is degree of freedom, the "MS" is the mean square, and the "F-ratio" is the F value which represent the extent of random error effect. It can be found from Table.5 that except for the Hs, all the p values between the two loading conditions are 0, implying a significant difference between them.

**Table.5. Statistical analysis of P value from two-sample t tests of the fully-loaded and ballast loading conditions.**

| Source | SS | df | MS | F-ratio | p-value |
|---|---|---|---|---|---|
| RPM | 5626.1446 | 1 | 5626.1446 | 421.9828 | 0 |
| SOG | 969.3455 | 1 | 969.3455 | 199.4155 | 0 |
| Pitch | 18.3941 | 1 | 18.3941 | 12.664 | 0.0004 |
| Roll | 888.4506 | 1 | 888.4506 | 53.6440 | 0 |
| Hs | 1.6426 | 1 | 1.6426 | 1.0687 | 0.3013 |
| MDS | 4344.3152 | 1 | 4344.3152 | 23.9842 | 0 |
| RWD | 683107.1025 | 1 | 683.17.1025 | 420.5264 | 0 |
| Wave Steepness | 0.0095 | 1 | 0.0095 | 163.8423 | 0 |

b.      *"The warning criteria in Figure 14 is the main results of this study, as I understand it, and I wonder if the is based on a weak foundation (a crude correlation analysis for four situations). Does this really push state-of-the art in weather warning criteria?"*

**Address to comment# B-b:**

It should be noticed here that the ship used for measurement is **a merchant ship (not a research ship)** which actually always tried to avoid rough seas, and compared with other studies using on-board measurement data (), we do not think our data is not enough to help improve the state-of-the art in weather routing literature. Besides, although the present data analysis only focus on four rough sea cases, the data was 10-min averaged and thus totally around 3000 data was analyzed for each ocean parameter (Hs, MDS, RWD, Wave Steepness) and ship response (RMP, Speed, Pitch, Roll), so **totally 24,000 data** was used with ranges of Hs from 0.705m to 5.69 m, MDS from 19.7 to 76.9 degree, RWD from 12.2 to 269 degree, Wave steepness from 0.006 to 0.0342, pitch motion from

0.27 to 6.32 degree, roll motion from 0.50 to 22 degree, RPM from 95 to 120, and speed from 2.46 to 14.6 knots. Additionally, **these four cases occurred in different ocean regions and time periods, which can be representative of rough sea navigation**.

Modifications in the conclusion part of resubmission are as follows:

Although only one bulk carrier has been studied in the present study, however, it should be noticed that according to the Office of Data and Economic Analysis from US Bureau of Economic Analysis (BEA), world's bulk carrier fleet includes 6,225 ships of over 10,000 DWT, and represent 40% of all ships in terms of tonnage and 39.4% in terms of vessels. Therefore, for bulk carriers of similar dimensions, these results can provide practical suggestions to ship operators on identifying and avoiding the possible high-risk ocean regions.

**Address to comment# C (journal selection):**

Address to comment "Hence, I don't find this paper to be interesting enough to recommend publication. On another note, I do not find NHESS to be the most relevant channel for presenting this research and would suggest that the authors possibly submit a revised version to a more ship-focused journal." is given as follows:

We submitted this study to the NHESS considering about two reasons, one is the scope of the NHESS given as follows:

*"the detection, monitoring, and modelling of natural phenomena, and the integration of measurements and models for the understanding and forecasting of the behaviour and the spatial and temporal evolution of hazardous natural events as well as their consequences;*

*"the design, development, experimentation, and validation of new techniques, methods, and tools for the detection, mapping, monitoring, and modelling of natural hazards and their human, environmental, and societal consequences;"*

Another reason is, as shown in the last reference, Zhang Zhiwei, 2017 published their study on "Global ship accidents and ocean swell-related sea states." in NHESS. Although our study has no direct accident data, but the large motion responses affected by weather conditions have a big potential leading to serve accidents, if not avoided immediately.

Thank you!

**Comment #2:**

The "warning criteria" are based on a rather simple statistical analysis (essentially only studying correlations between sea state variables and ship motion responses). They seem

to be very simplistic and I wonder if not state-of-the art numerical response calculations could be used to obtain better decision support. I guess you may easily estimate the effect of the different sea state variables on ship responses by performing a set of numerical simulations using hydrodynamic models etc (for any loading condition). How does your proposed method compare with such methods and what is the benefit of doing the simple correlation analysis to suggest warning criteria. However, I believe the collected dataset is interesting, and possibly this could be used to validate such numerical models (in addition to model tests, which are commonly used to validate ship model simulations).

**Address to comment #2:**

Thank you for your comment!

Regarding to your present comment, we added calculation of pitch motion by using hydrodynamic method EUT and NSM, with details shown as follows:

According to Nielsen (2008), ship motion measurements can be used to estimate the sea state even when high-frequency wave components of the wind wave spectrum are considered. Therefore, wave model results are validated using ship motion calculations. In our study, we calculated the pitch motion for wave validation owing to its close relationship with ship speed loss and other phenomena such as slamming, green water, and propeller racing, all of which are important for ship and cargo safety. To generate the pitch motion, we assume that ship motion is proportional to the directional wave spectrum:

$$D_p(\omega, \theta, V) = \frac{|X_p(\omega,\theta,V)|^2}{|1 - 2\omega_0 V \cos\theta / g|} D_W(\omega_0, \theta)$$

where $D_p(\omega, \theta, V)$ represents the directional pitch spectrum, ω is the encounter circular frequency, θ is the relative wave direction, V is the ship speed, D_W (ω_0,θ) is the directional wave spectrum (ω_0 is the circular frequency of incident waves), and X_p (ω,θ,V) is the response function of the pitch motion.

Among existing seakeeping models using potential theory and CFD, the enhanced unified theory (EUT; Kashiwagi, 1997) and the new strip method (NSM; Salvesen et al., 1970) were used for the calculation of the response function of pitch motion considering both the computational efficiency and accuracy, as shown in Fig. 4.

Moreover, ship motion was assumed following a Rayleigh distribution, which enables the calculation of the significant amplitude of pitch as follows:

$$P_{A_{1/3}} = 4.0 \sqrt{\int_0^{2\pi} \int_0^\infty D_p(\omega, \theta, V) d\omega d\theta}$$

For these 4 rough sea navigation cases, EUT and NSM were used to calculate pitch motion around the maximum pitch amplitude period for four different of wave modeling results: NCEP LINEAR, NCEP WRF, ERA LINEAR, and ERA WRF (shown as following figures).

Limitations of hydrodynamic models based on potential theory on the non-linearity can be seen from the differences among observation and calculations, thus, as a complement, we tried another way by using the direct statistical data analysis (*Data, Information, Knowledge, and Wisdom*) between sea state variables and ship motion responses.

[Figure]

1.      Kashiwagi, Masashi. "Numerical seakeeping calculations based on the slender ship theory." Ship Technology Research (Schiffstechnik) 4.4 (1997): 167-192.

2.    *Salvesen, Nils, E. O. Tuck, and Odd Faltinsen. "Ship motions and sea loads." Trans. SNAME 78.8 (1970): 250-287.*

Thank you!

**Comment #3:**

Some ships are equipped with a hull monitoring system to advise about operation in high seas. This is not mentioned in the paper, and a discussion on the effect of such systems, perhaps in combination with other methods for decision support could be relevant in the introduction or in section 2. Such systems measure stresses and accelerations in the ship hull and would provide additional information to what is collected in your case study. You could get ship structural response in addition to ship motion response that may be important in terms of safety and reliability.

**Address to comment #3:**

Thank you for your comment!

The hull monitoring system that measure stresses and accelerations in the ship hull is surely important in terms of decision support for ship safety and reliability, as you pointed out. However, such data of stresses and accelerations taken by hull monitoring system is not available this time. We hope we will have another ship installed with such monitoring system for measurement in the future. Anyway, references of the hull monitoring system have been added into the section 2, considering the important role it always plays, please confirm it.

Thank you!

**Comment #4:**

You study the correlation between sea state variables and ship response variables, and this is a major part of the paper. The times series are obviously highly autocorrelated. It is well known that strong serial correlation may give cross-correlated time-series even for independent variables, so the correlation coefficients given in Fig 6 should be interpreted with care. A note could be included on this (perhaps with reference). Moreover, what insight do you really get from all the correlations you find for the sea-states and ship response variables in sections 5.2/5.3? On p. 13 you state that the ocean waves will have a larger influence on ship motions in ballast conditions.
How can you conclude on that from higher correlation between roll and pitch?

**Address to comment #4:**

Thank you for your comment!

Reference has also been added in Sec.5.1 as:

Similarly, (Toffoli et al., 2005) also found that the reduction of spreading was observed to occur during growing sea state conditions for approximately 60% of the selected cases in his study on global ship accident analysis, as shown in Fig.7. And according to his results, the magnitude of the mean directional spread was found to decrease towards a value of 0.5 (approximately 25 degree, as shown in the top-left panel in Fig.6) with an enhancement of significant wave height, which also agree with our present study.

[Figure]

**Figure. 7. Correlation between the mean directional spread (σ) and the significant wave height: at the time of the maximum mean directional spread (upper panel) and at the time of the maximum significant wave height (lower panel). Accidents, for which the maximum spreading was recorded before the maximum significant wave heights are plotted as black diamonds (Toffoli et al., 2005).**

Besides, the wave height has a strong positive correlation coefficient with a wave steepness of 0.85, and the maximum value of wave steepness is 0.0342 in the present study, saying the ship in rough seas approached the limitation of accident, according to the study by (Toffoli et al., 2005) that pointed out that more than 50% of the incidents took place in sea states characterized by steepness larger than 0.035 (fully developed seas), as shown in Fig. 8.

[Figure]

**Figure. 8. Correlation plot of wave steepness and wave height: total sea (upper panel) and wind sea (lower panel). (Toffoli et al., 2005).**

Reference has also been added in Sec.5.2 as: "A relatively strong positive correlation can be found between the pitch and roll motion (0.660), ship speed and engine RPM (0.760), whereas a strong negative correlation is found between the pitch motion and ship speed (-0.854), almost agree with a previous study (0.883 for pitch motion and speed loss) by (Sasa, Kenji, et al., 2019) which focused on three cases using the same bulk carrier." and Fig. 9 in the resubmission.

[Figure]

**Figure. 9. Relation between the speed loss and the pitch motion (Sasa, Kenji, et al., 2019).**

Regarding to "Moreover, what insight do you really get from all the correlations you find for the sea-states and ship response variables in sections 5.2/5.3", we made 3 conclusions (first, second and third) in the section.6, which mainly include three parts (and these three parts are also the analysis process to obtain the final conclusion as shown

in Fig.14): "relationship among different **wave parameters** in actual rough seas", "relationship among the observed **ship responses**", as well as "relationship among the **ship responses to ocean states**".

The sentence "the ocean waves will have a larger influence on ship motions in ballast conditions." is imprecise, and it has been modified to in the resubmission: "a stronger correlation between roll and pitch motion can be found in the ballast (0.838) than that in the half-loaded cases (0.510), indicating a larger possibility of encountering head seas in loaded conditions, as shown in the top-left panel in Fig. 13."

Thank you!

**Comment #5:**

When you compare ship responses and what you refer to as navigation – can you assume that different behavior is only due to weather conditions? Is it the same crew operating the ship in all four situations? It is well known that different seafarers may respond differently, for example. Do you have control of other influencing factors (human factors etc.) For example, you state that ship operators usually reduces engine RPM more, but later, in half-loaded cases than in ballast conditions. Can you say this from just two cases of ballast and two cases in loaded conditions? Many other factors than the loading condition could be at play here.

**Address to comment #5:**

Thank you for your comment!

We agree with you that different seafarers may respond differently. To avoid misleading as well as make a more precise explanation, we have modified the imprecise sentence to: "In these focused 4 rough sea cases, ship operators tended to reduce the engine RPM more, but later in the half-loaded cases, than they did in the ballast ones."

We also agree with you that other factors could be play here, such as the human factors. And through the analysis using the regression models, we found low R-squared values of RPM and wave and ship parameters, and we made modifications in Section.5 the resubmission as follows:

"While other low "R-squared" values (marked as italic font numbers in Table. 7) give a low percent of variance explained. For instance, the fourth and last row giving the regression model of "RPM" against wave and ship motion parameters show a high possibility of influence of other crew operations (such as voluntary speed loss) on Engine RPM rather than the direct effects from ocean waves and wave-induced ship motion."

However, data of those factors are not collected this time and it could be another research field such as the crew psychology and praxeology, which is supposed to be taken into account when we have enough and deep investigation in the future. As the first step, here we want to make a research focusing on the loading condition, among all factors.

Additionally, to illustrate the limitations of our present study, we also made modifications in the end of conclusion the resubmission as follows:

"It should be also noticed that further investigations are necessary due to several limitations of the present research. Firstly, a larger number of ship navigation data (more ship types) and rough sea cases (more ocean regions) is supposed to be collected in the future to obtain a more convincing result. Then, other monitoring systems are needed to collect other important data such as the hull breaking due to the cumulative fatigue effects, as mentioned in the introduction part. Lastly, a deep insight of the influence of human factors on ship operations in rough sea will also be considered in the future study, considering its important role on a safe ship navigation."

Thank you!

**Minor comments:**

**Comment #6:**

"Due to an increase in ship size and number, shipping activities frequently lead to a higher possibility of ship accidents and increased safety risk to human beings, property losses, and the pollution of ocean environments.". I understand what you try to say, but this is poorly formulated. *: : ∴.* frequently lead to a higher possibility*: : :* What is meant by frequently? And higher than what?

**Address to comment #6:**

Thank you for your comment!

The sentence is imprecise, and we have modified it as: "…shipping activities lead to a high possibility of ship accidents…"

Thank you!

**Comment #7:**

Therefore, marine weather information, including an accurate forecast of extreme ocean surface wave states." This sentence has no verb and makes no sense

**Address to comment #7:**

Thank you for your comment!

The sentence has been modified to: "Therefore, an accurate forecast of extreme ocean surface wave states as well as the wave effects on ship navigation is essential for safe, economical, and environment-friendly ship navigation, from the viewpoint of ship weather routing."

Thank you!

**Comment #8:**

On p. 3 you mention different failure modes and potential problems related to waves. How about fatigue? I guess operation in severe weather can lead to increased fatigue on ship hull which may ultimately lead to failure. Fatigue is a cumulative effect and perhaps somewhat different from e.g. capsize and grounding, but it seems relevant to include here (possibly, with relevant references).

**Address to comment #8:**

Thank you for your comment!

We agree with what you pointed out here: the operation in severe weather can lead to increased fatigue on ship hull which may ultimately lead to failure. However, same as the case of hull monitoring system, such data of fatigue is not available this time, either. We hope we will have another ship installed with fatigue monitoring system for measurement of its cumulative effect in the future. Anyway, references of the hull monitoring system have been added into the introduction part, considering the important role it always plays, please confirm it.

Thank you!

**Comment #9:**

Why do you give two drafts for case 1 and 2 in Table 2?

**Address to comment #9:**

Thank you for your comment!

We give two drafts in case 1 and 2, which represent the "draft forward" and "draft aft" of the ship, respectively.

Thank you!

**Comment #10:**

On p.7 you mention 6 rough sea navigation cases. However, previously you only mention four cases. Check and update.

**Address to comment #10:**

Thank you for your comment!

The sentence has been modified to "A comparison of these three figures illustrates the high navigational risk of the selected 4 rough sea navigation cases, especially by the number 12, 13, and 18 provided in Fig. 3, which show a higher risk of experiencing unconventional waves."

Thank you!

**Comment #11:**

Your statement on p.8 should be backed up with a reference: "The most common sources of errors in wave model results are errors in the wind field". There are several other sources of errors. Can you validate this statement by a reference? Alternatively, just say "one of the most common errors..."

**Address to comment #11:**

Thank you for your comment!

The sentence has been modified to: "Meanwhile, one of the most common sources of errors in wave model results is the errors in the wind field."

Thank you!

**Comment #12:**

You state that wind input is important but use low-resolution wind input to drive the models. Will linear interpolation (in both space and time, or only in space? Do you assume six-hourly stationary conditions?) give accurate results? Could you use other downscaling methods (physical/statistical) to gain better results? Especially since you argue that the wind forcing is one of the most important sources of error of the numerical wave modelling.

**Address to comment #12:**

Thank you for your comment!

As the title of figure.5 shows, the WRF model has been used to downscale the GPV datasets of NCEP-FNL and ERA-Interim. It was a mistake we forgot putting the description of this method in the paper.

We have added the following sentences in the resubmission:

To drive WW3 using GPV datasets, the Weather Research and Forecasting (WRF) model (Skamarock, 2008) was used to generate the necessary near-surface wind fields. As a next-generation mesoscale numerical weather prediction system designed for both atmospheric research and operational forecasting applications, the WRF model has been widely used for typhoon simulations and real-time forecasting (Jianfeng et al., 2005; Davis et al., 2008; Cha and Wang., 2013).

Thank you!

**Comment #13:**

Abbreviations should be spelled out on first use. For example, what is WRF? (Weather Research and Forecast model??). MDS = Mean directional spreading? RWD = relative wave direction? (relative to wind? Relative to ship heading? OK, this is defined in Fig. 9, but not when it is first mentioned). RPM = revolutions per minute. GM = metacentric height (not obvious to all readers of this journal). Check throughout that all abbreviations are explained.

**Address to comment #13:**

Thank you for your comment!
All abbreviations have been spelled out on first use in the resubmission.
Thank you!

**Comment #14:**

What do you mean by "total" or "whole cases" on p. 12? I guess I understand what you mean (that you group statistics from all cases together?) but it should be better explained.

**Address to comment #14:**

Thank you for your comment!
The phase "whole cases" have been changed to "For the "Total" cases (Case 1, 2, 3, 4),…" in the resubmission.
Thank you!

**Comment #15:**

What do the colors in Figs 6, 7, etc. represent? Time? Value of one of the variables? This should be explained.

**Address to comment #15:**

Thank you for your comment!

We have added the sentence below the Fig. 6 as "The colors in these figures represent values of the variable of the vertical axis."

Thank you!

**Comment #16:**

Caption of Figs 6, 7,... should be revised. You are not showing correlations per se, but scatterplots of selected wave parameters to illustrate correlations. I believe you could be more precise.

**Address to comment #16:**

Thank you for your comment!

Captions of all related figures have been modified in the resubmission.

Thank you!

**Comment #17:**

This sentence on p. 13 does not make any sense: "As observed in the top-middle panel, as the pitch amplitude increases, the ship operators tend to further reduce the engine RPM (a higher correlation coefficient of -0.717), but later (when the pitch motion reaches approximately 3 degree in the half-loaded cases than in the ballast ones (-0.513 and less than 2 degree)." For one, the parentheses do not match. I can understand what you want to say, but you do not say it very well. Also, you continue to say that operators prefer to maintain speed, but the figure clearly shows that speed decreases as pitch increases. What do you try to say here?

**Address to comment #17:**

Thank you for your comment!

It was an imprecise expression, and the sentence has been modified as: "As observed in the top-middle panel, as the pitch amplitude increases, the ship operators tend to further reduce the engine RPM (a higher correlation coefficient of -0.717), but later (when the pitch motion reaches approximately 3 degree) in the half-loaded cases than in the ballast ones (a correlation coefficient of -0.513 and the pitch motion is less than 2 degree)."

Again, the sentence saying "operators prefer to maintain speed" has also been modified as: "As for the correlation between pitch motion and ship speed, as in the top-right panel, it is observed that the ship experienced similar speed loss in the beginning of the pitch motion increase (when the amplitude approaches 3 degree); while as the pitch motion increases from 3 to 6 degree, a larger and faster speed loss can be found in the half-loaded condition than that in the ballast one."

Thank you!

**Comment #18:**

- What do the dashed rings in Fig. 8, 9, ... represent? They are not discussed and should be removed (or discussed).

**Address to comment #18:**

Thank you for your comment!
All dashed rings have been removed.
Thank you!

**Comment #19:**

- On p. 20, the following sentence is very imprecise: "Ship responses such as… in ballast conditions are of an equal or slightly smaller amplitude than those in the halfloaded one". You are not comparing ship responses, but correlations between ship responses and sea states. Re-phrase to be more precise.

**Address to comment #19:**

Thank you for your comment!
The sentence has been modified to: "Correlations between sea states and ship responses such as the pitch motion (Fig.16-A ), engine RPM (Fig.16-C ) and ship speed (Fig.16-D ) in ballast conditions are of an equal or slightly smaller amplitude than those in the half-

loaded ones; while relatively large differences exist in the case of roll motion (Fig.16-B ).”

Thank you!

**Comment #20:**

- On page 22, values of Hs 5m, 4m and 1.5m are not the same as in Figure 14. Moreover, 3.5m, 3.8m and 1.3m is given in incorrect sequence (large, modest, small)

**Address to comment #20:**

Thank you for your comment!
It was a mistake, and we have modified it to the correct sequence in the resubmission.
Thank you!

**Comment #21:**

The presentation of the warning criteria in Fig. 14 is counter-intuitive, with large responses corresponding to small radius in the circle, and small responses far out with large radius. A minor issue, but strange that you chose to present it this way

**Address to comment #21:**

Thank you for your comment!
The idea of this figure is based on an imagination of putting the ship in the center (such as a typhoon center) of all circles, thus the large responses should exist in the circle closer to the center while the smaller responses occur in regions farther away from the center.
Thank you!

**Comment #22:**

- Reference list must be updated. List all authors not only et al. in the references list. Moreover, several citations in the text cannot be found in the reference list. E.g. Chen et al. 2013; 2015; 2018, ...

**Address to comment #22:**

Thank you for your comment!

Reference list has been updated with newly-added references as well as modifications of those "et al" ones.

Thank you!

Finally, authors want to express their appreciations to the reviewer for his/her precious time and efforts on my submission! I hope I can have more chances learning from you, if possible, in the future because I really think you can give me more valuable suggestions to my research in the future.

---

## Referee Comment (RC2) · Anonymous Referee #2 · 5 Apr 2020

The manuscript deals with an important problem. The chosen approach is to search for correlations among available datasets.

The manuscript is directed toward a very ship and nautical oriented audience. I encourage the authors to reconsider if the readership of NHESS is expected to have the appropriate ship orientation to appreciate the importance of this paper, and if the ship community is sufficiently aware of this journal for the paper to have the intended impact.

Following is a list of some minor issues:

[Figure]

1. Near line 25: This is not a complete sentence: "Therefore, marine weather information, including an accurate forecast of extreme ocean surface wave states."

2. Near line 35: There are many different ways to define steepness, leading to different numerical values. This is not a problem as long as the definitions are clearly stated. You refer to others, and you give your own values. Please state which definition is employed by your references and by yourself. In particular, did your references use the definition you suggest in Table 3?

3. Line 48: Substitute "serve" with "severe"?

4. In equations (2)-(4) it appears that $k$ is used both as vector and scalar? Please state what $k$ is, and please use different symbols for vectors and scalars. The symbol $d$ has not been defined. Please use parentheses around a product if a differential operator is supposed to act on the product.

5. In line 216 it is probably better to say $E$ is the variance of the surface elevation.

6. Near line 258 a closing parenthesis is lacking.

---

## Author Comment (AC2) · 6 Apr 2020

Dear Referee #2,

Thank you very much for your time on my submission! We have made addresses to your comments point by point, and we hope our addresses are satisfying and we are looking forward to your further comments, if any, to improve our manuscript. We have uploaded our addresses to your comment as a Supplement file named "Addresses to Comments, Reviewer 2". Thank you!

[Figure]

Please also note the supplement to this comment:
https://www.nat-hazards-earth-syst-sci-discuss.net/nhess-2019-399/nhess-2019-399-
AC2-supplement.pdf

―――――――――――――――――――――――
2019-399, 2020.

[Figure]

**Supplement:**

*Dear Anonymous Referee #2:*

*Thank you very much for your kind time on my submission!*

*I really appreciate your valuable and poignant comments on helping improving the present manuscript, and we have the following addresses to your comments. We hope our responses and modifications are satisfying.* *Modifications in the resubmission are given with a yellow-color background here in our addresses.*

*ChenChen*

**Comments #1**

**(1) Comments from Referees**

*The manuscript deals with an important problem. The chosen approach is to search for correlations among available datasets. The manuscript is directed toward a very ship and nautical oriented audience. I encourage the authors to reconsider if the readership of NHESS is expected to have the appropriate ship orientation to appreciate the importance of this paper, and if the ship community is sufficiently aware of this journal for the paper to have the intended impact.*

**(2) Author's response**

Thank you for your words recognizing the present study as an important problem! Regarding to your suggestion to reconsider if the NHESS is appreciate for publication of our study, we have the following considerations, which we hope can make it clear why we chose the NHESS.

**At first,** we submitted this study to the NHESS considering about the scopes of the NHESS given as follows, and we consider it appropriate to publish our study here since the NHESS did not refuse our submission directly due to the reason such as "out of scope of the NHESS".

"the detection, monitoring, and modelling of natural phenomena, and the integration of measurements and models for the understanding and forecasting of the behaviour and the spatial and temporal evolution of hazardous natural events as well as their consequences;

"the design, development, experimentation, and validation of new techniques, methods, and tools for the detection, mapping, monitoring, and modelling of natural hazards and their human, environmental, and societal consequences;"

**Secondly,** as you said the present study focuses on "a very ship and nautical" issue. But in our opinion, the differences between our study and most of traditional "very ship and nautical" studies are concluded as follows.

| | Research objectives | Research approaches |
|---|---|---|
| Traditional "very ship and nautical" studies | Optimum design of ship's hull, ship maneuvering (in waves), ship speed loss, ship stability (in wind and waves), ship structure safety (in waves), etc. | Marine hydrodynamics (potential theory or CFD methods), marine structural mechanics (potential theory or CFD methods), ship model test in a towing tank, etc. |
| Our present study | Application of geophysical fluid dynamics (wave simulation of extreme wave states) to increase marine safety by avoiding possible marine accidents induced by rough nature environment. | Geophysical fluid dynamics (wave simulation by CFD), statistical method (correlation and regression analysis). |

**Finally,** we consider it appreciate to be published on NHESS for two aspects. **One** is that it may also help the readership of NHESS to expand their horizons to contribute their experiences and knowledge of geosciences, which are usually lack of existence in researchers of ship field, to such important marine issues. **The other consideration** is that the ship community can also easily find this study by searching key words in scientific databases owning to the convenience of internet and various browsers at present, if they are willing to focus on a similar topic, just as what we have done in searching the study by Zhang Zhiwei, 2017.

**Comments #2**

**(1) Comments from Referees**

*Near line 25: This is not a complete sentence: "Therefore, marine weather information, including an accurate forecast of extreme ocean surface wave states."*

**(2) Author's response**

Thank you very much for your comments!

We have modified it in the resubmission to the sentence as shown in the following "Author's changes in manuscript".

**(3) Author's changes in manuscript.**

Therefore, an accurate forecast of extreme ocean surface wave states as well as the wave effects on ship navigation is essential for safe, economical, and environment-friendly ship navigation, from the viewpoint of ship weather routing.

**Comments #3**

**(1) Comments from Referees**

*Near line 35: There are many different ways to define steepness, leading to different numerical values. This is not a problem as long as the definitions are clearly stated. You refer to others, and you give your own values. Please state which definition is employed by your references and by yourself. In particular, did your references use the definition you suggest in Table 3?*

**(2) Author's response**

Thank you very much for your comments!

As you pointed out here, there are many different ways to define steepness, leading to different numerical values. And we have modified it in our resubmission as shown in the following "Author's changes in manuscript".

**(3) Author's changes in manuscript.**

It should be noticed here that the definition of wave steepness employed in the reference (Toffoli et al., 2005) is given as: $2\pi H_{m0} \big/ g T_{\mathrm{m}-10}^2$, where the definition of $H_{m0}$ by (Toffoli et al., 2005) also represents the significant wave height, as the definition of $H_s$; and the definition of $T_{\mathrm{m}-10}$ by (Toffoli et al., 2005) is the energy wave period: $T_{\mathrm{m}-10} = m_{-1} \big/ m_0$, where the $m_n$ is the nth-order moment of wave spectrum.

**Comments #4**

**(1) Comments from Referees**

*Line 48: Substitute "serve" with "severe"?*

**(2) Author's response**

Thank you very much for your comments!

We have modified it in the resubmission to the sentence as shown in the following "Author's changes in manuscript".

**(3) Author's changes in manuscript.**

From the ship accident statistics, they concluded that both the moderate, but rapid developing seas, as well as the seas more severe than the averaged local wave climate are closely related to the higher risk of ship accidents.

**Comments #5**

**(1) Comments from Referees**

*In equations (2)-(4) it appears that k is used both as vector and scalar? Please state what k is, and please use different symbols for vectors and scalars. The symbol d has not been defined. Please use parentheses around a product if a differential operator is supposed to act on the product.*

**(2) Author's response**

Thank you very much for your comments!

We think the reviewer is talking about equations (1)-(3), and we have modified them in the resubmission, as shown in the following "Author's changes in manuscript".

**(3) Author's changes in manuscript.**

$$\frac{\partial N}{\partial t} + \nabla_x \cdot \left(c_g + U\right)N + \frac{\partial}{\partial K}\widehat{K}N + \frac{\partial}{\partial \theta}\widehat{\theta}N = \frac{S}{\sigma} \tag{1}$$

$$\widehat{K} = -\frac{\partial \sigma}{\partial d}\frac{\partial d}{\partial s} - k \cdot \frac{\partial U}{\partial s} \tag{2}$$

$$\widehat{\theta} = -\frac{1}{K}\left(\frac{\partial \sigma}{\partial d}\frac{\partial d}{\partial m} + k \cdot \frac{\partial U}{\partial m}\right) \tag{3}$$

where N is the vector wavenumber spectrum, $c_g$ is the wave group velocity, $U$ is the current velocity, $s$ is the coordinate in the direction of $\theta$, d is the mean water depth, K is the wave number as a scalar, $k$ is the wavenumber vector, $m$ is the coordinate perpendicular to $s$, and $S$ is the net source term for the spectrum, $\sigma$ is the intrinsic

wave radian frequency.

**Comments #6**

**(1) Comments from Referees**

*In line 216 it is probably better to say E is the variance of the surface elevation.*

**(2) Author's response**

Thank you very much for your comments!

We have modified it in the resubmission to the sentence as shown in the following "Author's changes in manuscript".

**(3) Author's changes in manuscript.**

Here, the ==variance of the surface elevation== is $E = \int_0^{2\pi} \int_0^{\infty} F(f_r, \theta)\, df_r d\theta$, where $\sigma = 2\pi f_r$ is the intrinsic wave radian frequency, and $F(f_r, \theta)$ is the frequency-direction spectrum.

**Comments #7**

**(1) Comments from Referees**

*Near line 258 a closing parenthesis is lacking.*

**(2) Author's response**

Thank you very much for your comments!

We have modified it in the resubmission to the sentence as shown in the following "Author's changes in manuscript".

**(3) Author's changes in manuscript.**

As for the correlation between pitch motion and ship speed, as in the top-right panel, it is observed that the ship experienced similar speed loss in the beginning of the pitch motion increase (when the amplitude approaches 3 degree==);== while as the pitch motion increases from 3 to 6 degree, a larger and faster speed loss can be found in the half-loaded condition than that in the ballast one.

---

## Author Comment (AC3) · 6 Apr 2020

Dear Editor,

We have to express our appreciation to the editor for his efforts on our submission! The final-response form, uploaded as a supplement file named "Final-response form", is submitted to you for making a decision about the further handling of our manuscript. We hope our responses and modifications are satisfying. Modifications in the resubmission are given with a yellow-color background here in our addresses.

[Figure]
Thank you!

Please also note the supplement to this comment:
https://www.nat-hazards-earth-syst-sci-discuss.net/nhess-2019-399/nhess-2019-399-AC3-supplement.pdf

**Supplement:**

**Final-Response Form**

We have to express our appreciation to the editor for his efforts on our submission! This final-response form is submitted to the Editor for making a decision about the further handling of our manuscript. We hope our responses and modifications are satisfying. Modifications in the resubmission are given with a yellow-color background here in our addresses.

| Comments | Addresses |
|---|---|
| *This paper addresses the risk of operating ships in severe weather conditions and aims at providing decision support for ship navigation. It studies the relationship between weather conditions and motion responses of ships by comparing numerical wave modelling with on-board measurements of ship motions from a bulk carrier. This is an important issue for maritime safety, and also influence optimal weather routing that may lead to more cost-efficient and environmentally friendly shipping. Hence, the paper addresses an important issue for the maritime industry which is, in principle, relevant for publication.* | Thank you for your comment! We appreciate very much your kind words agreeing with the motivation of our study! We are here dividing this comment into three parts (**A:** language problem; **B:** data analysis; **C:** journal selection) and then making three part of addresses (shown as **A, B** and **C** as follows). **Address to comment# A (language problem):** Regarding to the poor language and presentation, as again pointed out in another minor comment "The language is at times poor, and thorough language vetting is recommended. The following are merely a few examples from the first parts of the paper but proof-reading by a native speaker is recommended throughout the paper", we are sorry about it but **we indeed used the English check service from the company "Editage"…** And we have revised the language in the resubmission. Please check it. |
| *However, I feel the quality of the paper is too poor to warrant publication in a quality international journal. The language and presentation of the research is poor and at times very* | **B: data analysis**
 a. Then, about your comment "*proposed warning criteria seems very simplistic and based on a very simple analysis of correlation between various variables*", we want to address it together with another two related comments: "*You study the correlation between sea states* |

*imprecise, and also the proposed warning criteria seems very simplistic and based on a very simple analysis of correlation between various variables. Hence, I don't find this paper to be interesting enough to recommend publication. On another note, I do not find NHESS to be the most relevant channel for presenting this research and would suggest that the authors possibly submit a revised version to a more ship-focused journal.*

*and ship responses. However, you only study one response/effect at a time. How do you account for interaction effects? That is, the effect of one parameter will be influenced by the value of another. How do you account for possible confounding effects? Could you gain insight if you try to fit a statistical regression model to these data, e.g. explaining the ship responses by the sea state variables. In such models you could include interaction terms to account for such dependencies and could perhaps give more insight than merely studying the (linear, I assume, but you do not say) correlation coefficients. Also, how statistically significant are the correlations you estimate? Particular with respect to section 5.4 and Fig 13 where you compare the correlations for loaded and ballast conditions this is a relevant question.*"

**Address to comment# B-a:**

To improve the paper as well as figure out the interaction terms to account for such dependencies as you said and recommended, a statistical regression model as well as statistically significances of these data has been added to explain the ship responses by the sea state variables in the end of Section.5.

Modifications in the resubmission are as follows:

Additionally, the interaction effects, meaning the effect of one parameter influenced by the value of another one, has also been studied using statistical regression model to further explain the relationship among ship responses and various sea state variables, as shown in Table. 7 (shown in the

Appendix in the end of the form).

Significant difference has also been added using a statistical method, the calculation of P value between two loading conditions, as shown in Table.5 in the resubmission. Results show that that except for the Hs, all the p values between the two loading conditions are 0, implying a significant difference between them.

Modifications in the resubmission are as follows:

To have a deep look at the difference between above-mentioned two loading conditions the significant test has been done, and the p values are given in Table. 5, where the "Source" is the parameter for which the significant test has been made, the "SS" is Sum of squares, the "df" is degree of freedom, the "MS" is the mean square, and the "F-ratio" is the F value which represent the extent of random error effect. It can be found from Table.5 that except for the Hs, all the p values between the two loading conditions are 0, implying a significant difference between them (shown in the Appendix in the end of the form).

b. *"The warning criteria in Figure 14 is the main results of this study, as I understand it, and I wonder if the is based on a weak foundation (a crude correlation analysis for four situations). Does this really push state-of-the art in weather warning criteria?"*

**Address to comment# B-b:**

It should be noticed here that the ship used for measurement is **a merchant ship (not a research ship)** which actually always tried to avoid rough seas, and compared with other studies using onboard measurement data (), we do not think our data is not enough to help improve the state-of-the art in weather routing literature. Besides, although the present data analysis only focus on four rough sea cases, the data was 10-min averaged and thus totally around 3000 data was analyzed for each ocean parameter (Hs, MDS, RWD, Wave Steepness) and ship response (RMP, Speed, Pitch, Roll), so **totally 24,000 data** was used with ranges of Hs from 0.705m to 5.69 m, MDS from 19.7 to 76.9 degree, RWD from 12.2 to 269 degree, Wave steepness from 0.006 to 0.0342, pitch motion from 0.27 to 6.32 degree, roll motion from 0.50 to 22 degree, RPM from 95 to 120, and speed from 2.46 to 14.6 knots. Additionally, **these four cases occurred in different ocean regions and time periods, which can be representative of rough sea navigation**.

Modifications in the conclusion part of resubmission are as follows:

Although only one bulk carrier has been studied in the present study, however, it should be noticed that according to the Office of Data and Economic Analysis from US Bureau of Economic Analysis (BEA), world's bulk carrier fleet includes 6,225 ships of over 10,000 DWT, and represent 40% of all ships in terms of tonnage and 39.4% in terms of vessels. Therefore, for bulk carriers of similar dimensions, these results can provide practical suggestions to ship operators on identifying and avoiding the possible high-risk ocean regions.

**Address to comment# C (journal selection):**
Address to comment "Hence, I don't find this paper to be interesting enough to recommend publication. On another note, I do not find NHESS

| | to be the most relevant channel for presenting this research and would suggest that the authors possibly submit a revised version to a more ship-focused journal." is given as follows: |
|---|---|
| | We submitted this study to the NHESS considering about two reasons, one is the scope of the NHESS given as follows: |
| | "the detection, monitoring, and modelling of natural phenomena, and the integration of measurements and models for the understanding and forecasting of the behaviour and the spatial and temporal evolution of hazardous natural events as well as their consequences; |
| | "the design, development, experimentation, and validation of new techniques, methods, and tools for the detection, mapping, monitoring, and modelling of natural hazards and their human, environmental, and societal consequences;" |
| | Another reason is, as shown in the last reference, Zhang Zhiwei, 2017 published their study on "Global ship accidents and ocean swell-related sea states." in NHESS. Although our study has no direct accident data, but the large motion responses affected by weather conditions have a big potential leading to serve accidents, if not avoided immediately. We can call this "To Nip Something in the Bud" instead of backward treatments of past accidents, if it could fit the scope of the NHESS. However, we agree with your comment that a more ship-focused journal may be also suitable. |
| | Thank you! |
| *The "warning criteria" are based on a rather simple statistical analysis (essentially only studying correlations between sea state variables and* | Thank you for your comment! Regarding to your present comment, we added calculation of pitch motion by using hydrodynamic method EUT and NSM, with details shown as follows: |

*ship motion responses). They seem to be very simplistic and I wonder if not state-of-the art numerical response calculations could be used to obtain better decision support. I guess you may easily estimate the effect of the different sea state variables on ship responses by performing a set of numerical simulations using hydrodynamic models etc (for any loading condition). How does your proposed method compare with such methods and what is the benefit of doing the simple correlation analysis to suggest warning criteria. However, I believe the collected dataset is interesting, and possibly this could be used to validate such numerical models (in addition to model tests, which are commonly used to validate ship model simulations).*

According to Nielsen (2008), ship motion measurements can be used to estimate the sea state even when high-frequency wave components of the wind wave spectrum are considered. Therefore, wave model results are validated using ship motion calculations. In our study, we calculated the pitch motion for wave validation owing to its close relationship with ship speed loss and other phenomena such as slamming, green water, and propeller racing, all of which are important for ship and cargo safety. To generate the pitch motion, we assume that ship motion is proportional to the directional wave spectrum.

$$D_p(\omega, \theta, V) = \frac{|X_p(\omega, \theta, V)|^2}{\left|1 - 2\omega_0 V \cos\theta / g\right|} D_W(\omega_0, \theta)$$

where $D_p(\omega, \theta, V)$ represents the directional pitch spectrum, $\omega$ is the encounter circular frequency, $\theta$ is the relative wave direction, V is the ship speed, $D\_W(\omega\_0, \theta)$ is the directional wave spectrum ($\omega\_0$ is the circular frequency of incident waves), and $X\_p(\omega, \theta, V)$ is the response function of the pitch motion.

Among existing seakeeping models using potential theory and CFD, the enhanced unified theory (EUT; Kashiwagi, 1997) and the new strip method (NSM; Salvesen et al., 1970) were used for the calculation of the response function of pitch motion considering both the computational efficiency and accuracy, as shown in **Fig. 4** (shown in the Appendix in the end of the form).

Moreover, ship motion was assumed following a Rayleigh distribution, which enables the calculation of the significant amplitude of pitch as follows.

$$P_{A_{1/3}} = 4.0 \sqrt{\int_0^{2\pi} \int_0^{\infty} D_p(\omega, \theta, V) d\omega d\theta}$$

| | For these 4 rough sea navigation cases, EUT and NSM were used to calculate pitch motion around the maximum pitch amplitude period for four different of wave modeling results: NCEP LINEAR, NCEP WRF, ERA LINEAR, and ERA WRF (shown as following figures). |
|---|---|
| | Limitations of hydrodynamic models based on potential theory on the non-linearity can be seen from the differences among observation and calculations, thus, as a complement, we tried another way by using the direct statistical data analysis (*Data, Information, Knowledge, and Wisdom*) between sea state variables and ship motion responses. |
| | **References:** |
| | 1. *Kashiwagi, Masashi. "Numerical seakeeping calculations based on the slender ship theory." Ship Technology Research (Schiffstechnik) 4.4 (1997): 167-192.* |
| | 2. *Salvesen, Nils, E. O. Tuck, and Odd Faltinsen. "Ship motions and sea loads."Trans. SNAME 78.8 (1970): 250-287.* |
| *Some ships are equipped with a hull monitoring system to advise about operation in high seas. This is not mentioned in the paper, and a discussion on the effect of such systems, perhaps in combination with other methods for decision support could be relevant in the introduction or in section 2. Such systems measure stresses and accelerations in the ship hull and would provide additional information to what is* | Thank you for your comment! |
| | The hull monitoring system that measure stresses and accelerations in the ship hull is surely important in terms of decision support for ship safety and reliability, as you pointed out. However, such data of stresses and accelerations taken by hull monitoring system is not available this time. We hope we will have another ship installed with such monitoring system for measurement in the future. Anyway, reference of the hull monitoring system have been added into the section 2, considering the important role it always plays, please confirm it. |
| | Thank you! |

| | |
|---|---|
| *collected in your case study. You could get ship structural response in addition to ship motion response that may be important in terms of safety and reliability.* | |
| *You study the correlation between sea state variables and ship response variables, and this is a major part of the paper. The times series are obviously highly autocorrelated. It is well known that strong serial correlation may give cross-correlated time-series even for independent variables, so the correlation coefficients given in Fig 6 should be interpreted with care. A note could be included on this (perhaps with reference). Moreover, what insight do you really get from all the correlations you find for the sea-states and ship response variables in sections 5.2/5.3? On p. 13 you state that the ocean waves will have a larger influence on ship motions in ballast conditions. How can you conclude on that from higher correlation between roll and pitch?* | Thank you for your comment! Reference has also been added in Sec.5.1 as: Similarly, (Toffoli et al., 2005) also found that the reduction of spreading was observed to occur during growing sea state conditions for approximately 60% of the selected cases in his study on global ship accident analysis, as shown in Fig.7. And according to his results, the magnitude of the mean directional spread was found to decrease towards a value of 0.5 (approximately 25 degree, as shown in the top-left panel in Fig.6) with an enhancement of significant wave height, which also agree with our present study.

Besides, the wave height has a strong positive correlation coefficient with a wave steepness of 0.85, and the maximum value of wave steepness is 0.0342 in the present study, saying the ship in rough seas approached the limitation of accident, according to the study by (Toffoli et al., 2005) that pointed out that more than 50% of the incidents took place in sea states characterized by steepness larger than 0.035 (fully developed seas), as shown in Fig. 8.

Reference has also been added in Sec.5.2 as: "A relatively strong positive correlation can be found between the pitch and roll motion (0.660), ship speed and engine RPM (0.760), whereas a strong negative correlation is found between the pitch motion and ship speed (-0.854), almost agree with a previous study (0.883 for pitch motion and speed |

loss) by (Sasa, Kenji, et al., 2019) which focused on three cases using the same bulk carrier." and Fig. 9 in the resubmission.

Regarding to "Moreover, what insight do you really get from all the correlations you find for the sea-states and ship response variables in sections 5.2/5.3", we made 3 conclusions (first, second and third) in the section.6, which mainly include three parts (and these three parts are also the analysis process to obtain the final conclusion as shown in Fig.14): "relationship among different **wave parameters** in actual rough seas", "relationship among the observed **ship responses**", as well as "relationship among the **ship responses to ocean states**".

The sentence "the ocean waves will have a larger influence on ship motions in ballast conditions." is imprecise, and it has been modified to in the resubmission: "a stronger correlation between roll and pitch motion can be found in the ballast (0.838) than that in the half-loaded cases (0.510), indicating a larger possibility of encountering head seas in loaded conditions, as shown in the top-left panel in Fig. 13."

Fig.7, 8 and 9 mentioned-above are given in the Appendix in the end of the form.

Thank you!

| | |
|---|---|
| *When you compare ship responses and what you refer to as navigation – can you assume that different behavior is only due to weather conditions? Is it the same crew operating the ship in all four situations? It is well known that different seafarers may respond differently, for* | Thank you for your comment!

 We agree with you that different seafarers may respond differently. To avoid misleading as well as make a more precise explanation, we have modified the imprecise sentence to: "In these focused 4 rough sea cases, ship operators tended to reduce the engine RPM more, but later in the half-loaded cases, than they did in the ballast ones."

 We also agree with you that other factors could |

| | |
|---|---|
| *example. Do you have control of other influencing factors (human factors etc.) For example, you state that ship operators usually reduces engine RPM more, but later, in half-loaded cases than in ballast conditions. Can you say this from just two cases of ballast and two cases in loaded conditions? Many other factors than the loading condition could be at play here.* | be play here, such as the human factors. However, data of those factors are not collected this time and it could be another research field such as the crew psychology and praxeology, which is supposed to be taken into account when we have enough and deep investigation in the future. As the first step, here we want to make a research focusing on the loading condition, among all factors.

Thank you! |
| *"Due to an increase in ship size and number, shipping activities frequently lead to a higher possibility of ship accidents and increased safety risk to human beings, property losses, and the pollution of ocean environments." I understand what you try to say, but this is poorly formulated... frequently lead to a higher possibility. What is meant by frequently? And higher than what?* | Thank you for your comment!

The sentence is imprecise, and we have modified it as: "…shipping activities lead to a high possibility of ship accidents…"

Thank you! |
| *Therefore, marine weather information, including an accurate forecast of extreme ocean surface wave states." This sentence has no verb and makes no sense.* | Thank you for your comment!

The sentence has been modified to: "Therefore, an accurate forecast of extreme ocean surface wave states as well as the wave effects on ship navigation is essential for safe, economical, and environment-friendly ship navigation, from the viewpoint of ship weather routing."

Thank you! |
| *On p. 3 you mention different failure modes and potential* | Thank you for your comment!

We agree with what you pointed out here: the |

| | |
|---|---|
| *problems related to waves. How about fatigue? I guess operation in severe weather can lead to increased fatigue on ship hull which may ultimately lead to failure. Fatigue is a cumulative effect and perhaps somewhat different from e.g. capsize and grounding, but it seems relevant to include here (possibly, with relevant references).* | operation in severe weather can lead to increased fatigue on ship hull which may ultimately lead to failure. However, same as the case of hull monitoring system, such data of fatigue is not available this time, either. We hope we will have another ship installed with fatigue monitoring system for measurement of its cumulative effect in the future. Anyway, reference of the hull monitoring system have been added into the introduction part, considering the important role it always plays, please confirm it. Thank you! |
| *Why do you give two drafts for case 1 and 2 in Table 2?* | Thank you for your comment! We give two drafts in case 1 and 2, which represent the "draft forward" and "draft aft" of the ship, respectively. Thank you! |
| *On p.7 you mention 6 rough sea navigation cases. However, previously you only mention four cases. Check and update.* | Thank you for your comment! The sentence has been modified to "A comparison of these three figures illustrates the high navigational risk of the selected 4 rough sea navigation cases, especially by the number 12, 13, and 18 provided in Fig. 3, which show a higher risk of experiencing unconventional waves." Thank you! |
| *Your statement on p.8 should be backed up with a reference: "The most common sources of errors in wave model results are errors in the wind field". There are several other sources of errors. Can you validate this statement by a reference? Alternatively, just say "one of the most common errors…"* | Thank you for your comment! The sentences has been modified to: "Meanwhile, one of the most common sources of errors in wave model results is the errors in the wind field." Thank you! |
| *You state that wind input is* | Thank you for your comment! |

| | |
|---|---|
| *important but use low-resolution wind input to drive the models. Will linear interpolation (in both space and time, or only in space? Do you assume six-hourly stationary conditions?) give accurate results? Could you use other downscaling methods (physical/statistical) to gain better results? Especially since you argue that the wind forcing is one of the most important sources of error of the numerical wave modelling.* | As the title of figure.5 shows, the WRF model has been used to downscale the GPV datasets of NCEP-FNL and ERA-Interim. It was a mistake we forgot putting the description of this method in the paper. We have added the following sentences in the resubmission: To drive WW3 using GPV datasets, the Weather Research and Forecasting (WRF) model (Skamarock, 2008) was used to generate the necessary near-surface wind fields. As a next-generation mesoscale numerical weather prediction system designed for both atmospheric research and operational forecasting applications, the WRF model has been widely used for typhoon simulations and real-time forecasting (Jianfeng et al., 2005; Davis et al., 2008; Cha and Wang., 2013). Thank you! |
| *Abbreviations should be spelled out on first use. For example, what is WRF? (Weather Research and Forecast model??). MDS = Mean directional spreading? RWD = relative wave direction? (relative to wind? Relative to ship heading? OK, this is defined in Fig. 9, but not when it is first mentioned). RPM = revolutions per minute. GM = metacentric height (not obvious to all readers of this journal). Check throughout that all abbreviations are explained.* | Thank you for your comment! All abbreviations have been spelled out on first use in the resubmission. Thank you! |
| *What do you mean by "total" or "whole cases" on p. 12? I guess* | Thank you for your comment! The phase "whole cases" have been changed to |

| | |
|---|---|
| *I understand what you mean (that you group statistics from all cases together?) but it should be better explained.* | "For the "Total" cases (Case 1, 2, 3, 4),…" in the resubmission.

 Thank you! |
| *What do the colors in Figs 6, 7, etc. represent? Time? Value of one of the variables? This should be explained.* | Thank you for your comment!

 We have added the sentence below the Fig. 6 as "The colors in these figures represent values of the variable of the vertical axis."

 Thank you! |
| *Caption of Figs 6, 7,… should be revised. You are not showing correlations per se, but scatterplots of selected wave parameters to illustrate correlations. I believe you could be more precise.* | Thank you for your comment!

 Captions of all related figures have been modified in the resubmission.

 Thank you! |
| *This sentence on p. 13 does not make any sense: "As observed in the top-middle panel, as the pitch amplitude increases, the ship operators tend to further reduce the engine RPM (a higher correlation coefficient of -0.717), but later (when the pitch motion reaches approximately 3 degree in the half-loaded cases than in the ballast ones (-0.513 and less than 2 degree)." For one, the parentheses do not match. I can understand what you want to say, but you do not say it very well. Also, you continue to say that operators prefer to maintain speed, but the figure clearly shows that speed decreases as pitch increases. What do you try* | Thank you for your comment!

 It was an imprecise expression, and the sentence has been modified as: "As observed in the top-middle panel, as the pitch amplitude increases, the ship operators tend to further reduce the engine RPM (a higher correlation coefficient of -0.717), but later (when the pitch motion reaches approximately 3 degree) in the half-loaded cases than in the ballast ones (a correlation coefficient of -0.513 and the pitch motion is less than 2 degree)."

 Again, the sentence saying "operators prefer to maintain speed" has also been modified as: "As for the correlation between pitch motion and ship speed, as in the top-right panel, it is observed that the ship experienced similar speed loss in the beginning of the pitch motion increase (when the amplitude approaches 3 degree); while as the pitch motion increases from 3 to 6 degree, a larger and faster speed loss can be found in the half-loaded condition than that in the ballast one." |

| | |
|---|---|
| *to say here?* | Thank you! |
| *- What do the dashed rings in Fig. 8, 9, … represent? They are not discussed and should be removed (or discussed).* | Thank you for your comment!
All dashed rings have been removed.
Thank you! |
| *- On p. 20, the following sentence is very imprecise: "Ship responses such as… in ballast conditions are of an equal or slightly smaller amplitude than those in the halfloaded one". You are not comparing ship responses, but correlations between ship responses and sea states. Re-phrase to be more precise.* | Thank you for your comment!
The sentence has been modified to: "Correlations between sea states and ship responses such as the pitch motion (Fig.16-A ), engine RPM (Fig.16-C ) and ship speed (Fig.16-D ) in ballast conditions are of an equal or slightly smaller amplitude than those in the half-loaded ones; while relatively large differences exist in the case of roll motion (Fig.16-B )."
Thank you! |
| *- On page 22, values of Hs 5m, 4m and 1.5m are not the same as in Figure 14. Moreover, 3.5m, 3.8m and 1.3m is given in incorrect sequence (large, modest, small)* | Thank you for your comment!
It was a mistake, and we have modified it to the correct sequence in the resubmission in correct sequence.
Thank you! |
| *The presentation of the warning criteria in Fig. 14 is counter-intuitive, with large responses corresponding to small radius in the circle, and small responses far out with large radius. A minor issue, but strange that you chose to present it this way* | Thank you for your comment!
The idea of this figure is based on an imagination of putting the ship in the center (such as a typhoon center) of all circles, thus the large responses should exist in the circle closer to the center while the smaller responses occur in regions farther away from the center.
Thank you! |
| *- Reference list must be updated. List all authors not only et al. in the references list. Moreover, several citations in the text cannot be found in the reference list. E.g. Chen et al. 2013; 2015;* | Thank you for your comment!
Reference list has been updated with newly-added references as well as modifications of those "et al" ones.
Thank you! |

| | |
|---|---|
| *2018, ...* | |
| *The manuscript deals with an important problem. The chosen approach is to search for correlations among available datasets. The manuscript is directed toward a very ship and nautical oriented audience. I encourage the authors to reconsider if the readership of NHESS is expected to have the appropriate ship orientation to appreciate the importance of this paper, and if the ship community is sufficiently aware of this journal for the paper to have the intended impact.* | Thank you for your words recognizing the present study as an important problem! Regarding to your suggestion to reconsider if the NHESS is appreciate for publication of our study, we have the following considerations, which we hope can make it clear why we chose the NHESS. **At first,** we submitted this study to the NHESS considering about the scopes of the NHESS given as follows, and we consider it appropriate to publish our study here since the NHESS did not refuse our submission directly due to the reason such as "out of scope of the NHESS". "the detection, monitoring, and modelling of natural phenomena, and the integration of measurements and models for the understanding and forecasting of the behaviour and the spatial and temporal evolution of hazardous natural events as well as their consequences; "the design, development, experimentation, and validation of new techniques, methods, and tools for the detection, mapping, monitoring, and modelling of natural hazards and their human, environmental, and societal consequences;" **Secondly,** as you said the present study focuses on "a very ship and nautical" issue. But in our opinion, the differences between our study and most of traditional "very ship and nautical" studies are concluded as Table.8 given in the Appendix. **Finally,** we consider it appreciate to be published on NHESS for two aspects. **One** is that it may also help the readership of NHESS to expand their horizons to contribute their experiences and knowledge of geosciences, which are usually lack of existence in researchers of ship field, to such |

| | important marine issues. **The other consideration** is that the ship community can also easily find this study by searching key words in scientific databases owning to the convenience of internet and various browsers at present, if they are willing to focus on a similar topic, just as what we have done in searching the study by Zhang Zhiwei, 2017. Thank you! |
|---|---|
| *Near line 25: This is not a complete sentence: "Therefore, marine weather information, including an accurate forecast of extreme ocean surface wave states."* | Thank you very much for your comments! We have modified it in the resubmission to the sentence as "Therefore, an accurate forecast of extreme ocean surface wave states as well as the wave effects on ship navigation is essential for safe, economical, and environment-friendly ship navigation, from the viewpoint of ship weather routing." Thank you! |
| *Near line 35: There are many different ways to define steepness, leading to different numerical values. This is not a problem as long as the definitions are clearly stated. You refer to others, and you give your own values. Please state which definition is employed by your references and by yourself. In particular, did your references use the definition you suggest in Table 3?* | Thank you very much for your comments! As you pointed out here, there are many different ways to define steepness, leading to different numerical values. And we have added it in our resubmission as "It should be noticed here that the definition of wave steepness employed in the reference (Toffoli et al., 2005) is given as: $2\pi H_{m0} \big/ g T_{\mathrm{m}-10}^{2}$, where the definition of $H_{m0}$ by (Toffoli et al., 2005) also represents the significant wave height, as the definition of $H_s$; and the definition of $T_{\mathrm{m}-10}$ by (Toffoli et al., 2005) is the energy wave period: $T_{\mathrm{m}-10} = m_{-1} \big/ m_0$, where the $m_n$ is the nth-order moment of wave spectrum." Thank you! |
| *Line 48: Substitute "serve" with "severe"?* | Thank you very much for your comments! We have modified it in the resubmission to the sentence "From the ship accident statistics, they |

| | |
|---|---|
| | concluded that both the moderate, but rapid developing seas, as well as the seas more severe than the averaged local wave climate are closely related to the higher risk of ship accidents." Thank you! |
| *In equations (2)-(4) it appears that k is used both as vector and scalar? Please state what k is, and please use different symbols for vectors and scalars. The symbol d has not been defined. Please use parentheses around a product if a differential operator is supposed to act on the product.* | Thank you very much for your comments! We think the reviewer is talking about equations (1)-(3), and we have modified them in the resubmission, shown as follows. $$\frac{\partial N}{\partial t} + \nabla_x \cdot \left(c_g + U\right)N + \frac{\partial}{\partial K}\widehat{K}N + \frac{\partial}{\partial \theta}\widehat{\theta}N = \frac{S}{\sigma}$$ (1) $$\widehat{K} = -\frac{\partial \sigma}{\partial d}\frac{\partial d}{\partial s} - k \cdot \frac{\partial U}{\partial s}$$ (2) $$\widehat{\theta} = -\frac{1}{K}\left(\frac{\partial \sigma}{\partial d}\frac{\partial d}{\partial m} + k \cdot \frac{\partial U}{\partial m}\right)$$ (3) where N is the vector wavenumber spectrum, $c_g$ is the wave group velocity, $U$ is the current velocity, $s$ is the coordinate in the direction of θ, d is the mean water depth, K is the wave number as a scalar, $k$ is the wavenumber vector, $m$ is the coordinate perpendicular to $s$, and $S$ is the net source term for the spectrum, $\sigma$ is the intrinsic wave radian frequency. |
| *In line 216 it is probably better to say E is the variance of the surface elevation.* | Thank you very much for your comments! We have modified it in the resubmission to the sentence as: "Here, the variance of the surface elevation is $E = \int_0^{2\pi}\int_0^{\infty} F(f_r, \theta)\, df_r d\theta$, where $\sigma = 2\pi f_r$ is the intrinsic wave radian frequency, and $F(f_r, \theta)$ is the frequency-direction spectrum." |
| *Near line 258 a closing parenthesis is lacking.* | Thank you very much for your comments! We have modified it in the resubmission to the |

| | sentence as: "As for the correlation between pitch motion and ship speed, as in the top-right panel, it is observed that the ship experienced similar speed loss in the beginning of the pitch motion increase (when the amplitude approaches 3 degree); while as the pitch motion increases from 3 to 6 degree, a larger and faster speed loss can be found in the half-loaded condition than that in the ballast one." |
| --- | --- |

**Appendix:**

Table.5. Statistical analysis of P value from two-sample t tests of the fully-loaded and ballast loading conditions.

| Source | SS | df | MS | F-ratio | p-value |
|--------|------|----|------|---------|---------|
| RPM | 5626.1446 | 1 | 5626.1446 | 421.9828 | 0 |
| SOG | 969.3455 | 1 | 969.3455 | 199.4155 | 0 |
| Pitch | 18.3941 | 1 | 18.3941 | 12.664 | 0.0004 |
| Roll | 888.4506 | 1 | 888.4506 | 53.6440 | 0 |
| Hs | 1.6426 | 1 | 1.6426 | 1.0687 | 0.3013 |
| MDS | 4344.3152 | 1 | 4344.3152 | 23.9842 | 0 |
| RWD | 683107.1025 | 1 | 683.17.1025 | 420.5264 | 0 |
| Wave Steepness | 0.0095 | 1 | 0.0095 | 163.8423 | 0 |

Table. 7. Relationship among ship responses and various sea state variables by statistical regression.

| |
|---|
| $Pitch = 0.3023 + 0.5877 * Hs - 0.0032 * MDS + 32.7620 * WS - 0.0055 * RWD$ |
| $Roll = 1.5923 + 2.3377 * Hs - 0.0343 * MDS + 0.0112 * RWD$ |
| $SOG = 13.6654 - 0.1427 * Hs - 175.1856 * WS + 0.0178 * RWD$ |
| $RPM = 108.8538 - 1.0308 * Hs + 0.0140 * MDS - 65.7744 * WS - 0.0075 * RWD$ |
| $SOG = 14.1316 - 1.9723 * Pitch + 0.1713 * Roll$ |
| $RPM = 109.6030 - 1.7790 * Pitch$ |

Table. 8. Differences between our study and most of traditional "very ship and nautical" studies

|  | Research objectives | Research approaches |
|---|---|---|
| Traditional "very ship and nautical" studies | Optimum design of ship's hull, ship maneuvering (in waves), ship speed loss, ship stability (in wind and waves), ship structure safety (in waves), etc. | Marine hydrodynamics (potential theory or CFD methods), marine structural mechanics (potential theory or CFD methods), ship model test in a towing tank, etc. |
| Our present study | Application of geophysical fluid dynamics (wave simulation of extreme wave states) to increase marine safety by avoiding possible marine accidents induced by rough nature environment. | Geophysical fluid dynamics (wave simulation by CFD), statistical method (correlation and regression analysis). |

[Figure]

Fig.4. Ship's pitch motion calculated by existing seakeeping models using potential theory and CFD, the enhanced unified theory (EUT; Kashiwagi, 1997) and the new strip method (NSM; Salvesen et al., 1970).

[Figure]

Figure. 7. Correlation between the mean directional spread (σ) and the significant wave height: at the time of the maximum mean directional spread (upper panel) and at the time of the maximum significant wave height (lower panel). Accidents, for which the maximum spreading was recorded before the maximum significant wave heights are plotted as black diamonds (Toffoli et al., 2005).

[Figure]

Figure. 8. Correlation plot of wave steepness and wave height: total sea (upper panel) and wind sea (lower panel). (Toffoli et al., 2005).

[Figure]

Figure. 9. Relation between the speed loss and the pitch motion (Sasa, Kenji, et al., 2019).